

# Key drivers of ozone change and its radiative forcing over the 21st century

**Fernando Iglesias-Suarez[1,2], Douglas E. Kinnison[3], Alexandru Rap[4],**

**Oliver Wild[1,2] and Paul J. Young[1,2,5]**

[1]Lancaster Environment Centre, Lancaster University, Lancaster, UK

[2]Data Science Institute, Lancaster University, Lancaster, UK

[3]Atmospheric Chemistry Observations and Modeling Laboratory, National Center for Atmospheric Research, Boulder, Colorado, USA

[4]School of Earth and Environment, University of Leeds, Leeds, UK

[5]Pentland Centre for Sustainability in Business, Lancaster University, Lancaster, UK

Correspondence to: F. Iglesias-Suarez (n.iglesiassuarez@lancaster.ac.uk)

## Abstract

Over the 21st century changes in both tropospheric and stratospheric ozone are likely to have important consequences for the Earth's radiative balance. In this study we investigated the radiative effects of future ozone changes, using the Community Earth System Model (CESM1), with the Whole Atmosphere Community Climate Model (WACCM), and including fully coupled radiation and chemistry schemes. Using year 20 2100 conditions from the Representative Concentration Pathways 8.5 (RCP8.5) scenario, we quantified the individual contributions to ozone radiative forcing of (1) climate change (with and without lightning feedback), (2) reduced concentrations of ozone depleting substances (ODSs), and (3) methane increases. We calculated future ozone radiative forcing relative to year 2000 of (1) $63 \pm 76$ mWm$^{-2}$, (2) 25 $129 \pm 81$ mWm$^{-2}$, and (3) $225 \pm 85$ mWm$^{-2}$, due to climate change, ODSs and methane respectively. Our best estimate of net ozone forcing in this set of simulations is $420 \pm 120$ mWm$^{-2}$ relative to year 2000, and $750 \pm 230$ mWm$^{-2}$ relative to year 1750, with uncertainty range given by approximately $\pm 30$ %. We find that the overall long-term tropospheric ozone forcing from methane chemistry-climate feedbacks




related to OH and methane lifetime is small (46 mWm$^{-2}$). Ozone forcings associated with climate change and stratospheric ozone recovery are robust with regard to background conditions, even though the ozone response is sensitive to both changes in atmospheric composition and climate. Changes in stratospheric-produced ozone account for ~47 % of the overall radiative forcing in this set of simulations, highlighting the key role of the stratosphere in determining future radiative forcing.

## 1    Introduction

Ozone is an important trace gas that plays a key role in the Earth's radiative budget, atmospheric chemistry and air quality. As a radiatively active species, ozone interacts with both shortwave and longwave radiation. In the troposphere, ozone is an important regulator of the oxidising capacity (both itself and as the main source of hydroxyl radicals, OH), as well as being an important pollutant, with negative effects on vegetation and human health (e.g. Prather et al., 2001; UNEP, 2015). However, approximately 90% of ozone is found in the stratosphere – protecting the biosphere from harmful ultraviolet solar radiation (WMO, 2014) – and is an important source of ozone in the troposphere and its budget (e.g. Collins et al., 2003; Sudo et al., 2003; Zeng and Pyle, 2003). Therefore, its future evolution – in the troposphere and the stratosphere – is a main concern for climate change and air quality during the 21st century. Future changes in emissions of ozone precursors (e.g. methane), ozone depleting substances (ODSs) and climate are thought to be major drivers of ozone abundances (e.g. Stevenson et al., 2006; Kawase et al., 2011; Young et al., 2013; Banerjee et al., 2016).

Stratospheric-tropospheric exchange (STE) of ozone significantly influences the abundance and distribution of tropospheric ozone (e.g. Banerjee et al., 2016). Although, observational estimates and climate models suggest an acceleration of the stratospheric mean mass transport via the Brewer-Dobson circulation (BDC) associated with climate change (e.g. Oberländer et al., 2013; Ploeger et al., 2013; Butchart, 2014; Stiller et al., 2017), significant uncertainty still remains (Engel et al., 2009; Hegglin et al., 2014; Ray et al., 2014). The BDC is the wave-driven factor governing the transport of air and trace constituents in the stratosphere, and characterized by upwelling in the tropics, poleward motion in the stratosphere and

sinking at middle and high latitudes (Butchart, 2014, and references therein). The BDC is commonly thought to consist of a shallow branch, controlling the lower stratosphere region, and a deep branch controlling the middle and upper stratosphere. The latter presents two cells during the spring and fall seasons, and one stronger cell

into the winter hemisphere (Birner and Bönisch, 2011). The tropopause is the boundary that "separates" the troposphere and the stratosphere, two chemically and dynamically distinct regions. Defining the tropopause is crucial to diagnose budget terms of trace gases such as the STE of ozone (e.g. Prather et al., 2011), although the chosen definition may affect the resulting analysis (e.g. Wild, 2007; Stevenson et al.,

2013; Young et al., 2013).

Stratospheric ozone is expected to recover to pre-industrial levels during the 21st century due to the implementation of the Montreal Protocol and its Amendments and Adjustments (WMO, 2014), as ODSs concentrations slowly decrease in the atmosphere (e.g. Austin and Wilson, 2006; Eyring et al., 2010). Indeed, the global

ozone layer has already shown the first signs of recovery (WMO, 2014; Chipperfield et al., 2017). Future ozone recovery can affect tropospheric composition via enhanced STE of ozone and reductions in tropospheric photolysis rates, both associated with higher levels of ozone in the stratosphere. Previous modelling studies that have isolated the impacts of stratospheric ozone recovery have shown that the increased

STE is the most important driver of changes in the tropospheric ozone burden (Zeng et al., 2010; Kawase et al., 2011; Banerjee et al., 2016). However, reductions in ozone photolysis result in lower OH concentrations – i.e. $O_3 + h\nu\ (\lambda < 320\ \text{nm}) \rightarrow O(^1D) + O_2$ – and therefore longer methane lifetime (e.g. Morgenstern et al., 2013; Zhang et al., 2014).

Impacts of future climate change on ozone are robust across a number of modelling studies and multi-model activities (Kawase et al., 2011; Young et al., 2013; Arblaster et al., 2014; Banerjee et al., 2016; Iglesias-Suarez et al., 2016). Stratospheric cooling leads to further ozone loss in the lower stratosphere (through enhanced heterogeneous ozone destruction) and ozone increases in the upper

stratosphere (through reduced $NO_x$ abundances and $HO_x$-catalysed ozone loss, and enhanced net oxygen chemistry) (Haigh and Pyle, 1982; Rosenfield et al., 2002). In addition, a projected acceleration of the BDC leads to an enhanced STE of ozone (e.g. Garcia and Randel, 2008; Butchart et al., 2010), which results in (i) decreases in





tropical lower stratospheric ozone, associated to an increase of relatively ozone-poor air entering from the troposphere; and (ii) ozone increases in the upper troposphere, particularly in the region of the subtropical jets, linked to the descending branch of the BDC. On the other hand, a warmer and wetter climate results in lower tropospheric

ozone levels – i.e. linked to a decrease in net chemical production due to enhanced ozone chemical loss – (e.g. Wild, 2007).

Climate feedbacks associated with future ozone changes are surrounded by large uncertainties. Lightning is a major natural source of nitrogen dioxides ($LNO_x$) in the troposphere and the only source away from the surface (Galloway et al., 2004),

with important consequences for atmospheric composition in the mid-upper troposphere and the lower stratosphere. The current best estimate of annual and global mean $LNO_x$ emissions is $5 \pm 3$ Tg(N) $yr^{-1}$, with chemistry-climate models suggesting $LNO_x$ emissions sensitivity to climate change of ~ 4–60 % $K^{-1}$ (Schumann and Huntrieser, 2007). Although more recent modelling studies find $LNO_x$ emissions

climate sensitivity lying at the lower end of the above estimate (Zeng et al., 2008; Banerjee et al., 2014), results from a multi-model activity suggest large uncertainty in the magnitude and even the sign of future projections response due to different parameterizations (Finney et al., 2016). Most $LNO_x$ emissions occur in the mid-upper tropical troposphere over the continents, where photochemical production of ozone is

most efficient in the troposphere – i.e. low background concentrations and longer lifetimes of $NO_x$, lower temperatures affecting ozone loss chemistry and abundant sunlight (e.g. Williams, 2005; Dahlmann et al., 2011). A small but significant fraction of lightning-induced $NO_x$ emissions are converted into less photochemically active nitric acid ($HNO_3$, via $HO_2 + NO$ reaction), which can be removed through wet

deposition or transported into the lower stratosphere (acting as a reservoir of $NO_x$) (e.g. Jacob, 1999; Søvde et al., 2011). In addition, OH concentrations increase with $LNO_x$ emissions and the resultant lightning-produced ozone – i.e. via $NO + HO_2$ and $O(^1D) + H_2O$ respectively – with a corresponding reduction in methane lifetime. This resulting climate feedback is important because methane is a potent GHG and ozone

precursor.

To date, ozone is the third largest contributor to the total radiative forcing (RF) since the pre-industrial period, with its overall increases contributing $+0.35$ $Wm^{-2}$ (Myhre et al., 2013). We use the concept of radiative effect (RE) to



diagnose the impact of ozone abundances and distributions on the atmospheric radiative budget. The ozone RE is the radiative flux imbalance between incoming shortwave solar radiation and outgoing longwave infrared radiation (at the tropopause, after allowing for stratospheric temperatures to readjust to radiative

equilibrium) which results from the presence of both anthropogenic and natural ozone (Rap et al., 2015). Note that RF is therefore the change in RE over time (e.g. Myhre et al., 2013). Ozone shows two distinct regimes with regard to its RE, with positive (longwave radiation) and negative (shortwave radiation) effects for increases in stratospheric ozone, and positive (for both longwave and shortwave radiation) effects

for ozone increases in the troposphere (e.g. Lacis et al., 1990; Forster and Shine, 1997). In addition, changes in ozone distribution – i.e. latitudinal and vertical structure – are of a particular interest for its RE, due to horizontally varying factors such as, surface albedo, clouds and the thermal structure of the atmosphere (e.g. Lacis et al., 1990; Berntsen et al., 1997; Forster and Shine, 1997; Gauss et al., 2003).

Previous studies showed highest radiative efficiency of ozone in the tropical upper troposphere (e.g. Worden et al., 2011; Riese et al., 2012; Rap et al., 2015), a region greatly influenced by stratospheric influx (e.g. Hegglin and Shepherd, 2009; Zeng et al., 2010) and lightning-produced ozone (e.g. Liaskos et al., 2015).

Modelling experiments used in the latest Assessment Report of the
Intergovernmental Panel on Climate Change (IPCC) followed the Representative Concentration Pathways (RCPs) emission scenarios for short-lived precursors (van Vuuren et al., 2011) and long-lived species (Meinshausen et al., 2011). The RCPs are named according to the total radiative forcing at the end of the 21st century relative to 1750. For example, the RCP8.5 emission scenario refers to the total 8.5 $Wm^{-2}$ RF by
2100. To different extents, all RCPs adopt future global stringent air quality regulations by 2100 relative to year 2000, resulting in reductions of $NO_x$ and Non Methane Volatile Organic Compounds (NMVOCs) emissions. However, methane emissions decrease in all RCPs but for RCP8.5 – i.e. more than doubled emissions are assumed – for the same period.

Previous research has investigated impacts on ozone abundances and distributions associated to future changes in climate, ODSs and ozone precursor emissions in a processed-based approach (Collins et al., 2003; Sudo et al., 2003; Zeng and Pyle, 2003; Zeng et al., 2008; Zeng et al., 2010; Kawase et al., 2011; Banerjee et





al., 2016). Other modelling studies focused on the radiative effects of mainly tropospheric ozone changes under future emission scenarios in a non processed-based fashion (e.g. Gauss et al., 2003; Stevenson et al., 2013). One study has recently identified the indirect tropospheric and stratospheric ozone RF between 2000 and
2100 due to individual perturbations (Banerjee et al., 2017). Yet the upper limit of future ozone RF remains poorly constrained. For example, climate models do not even necessarily agree on the sign of the indirect ozone forcing resulting from climate change and associated feedbacks (i.e. $LNO_x$). Furthermore, there are uncertainties arising from the interactions and non-linearities between different agents (i.e.
combined forcing may differ from the sum of individual forcings due to different background conditions), as well as and long-term changes (i.e. methane feedback associated with changes in lifetimes).

Here we aim to narrow this gap by assessing how key factors drive net ozone radiative forcing, and provide a gauge of the uncertainty arising from non-linearities
and long-term feedbacks. We use the Community Earth System Model (CESM1) in its "high-top" (up to 140 km) atmosphere version – the Whole Atmosphere Community Climate Model (WACCM) – and a series of sensitivity simulations to quantify the radiative effects of ozone due to future climate change, stratospheric ozone recovery, and methane emissions between 2000 and 2100. Particular attention
is given to non-linearities emerging from climate change and ozone recovery, lightning-produced ozone (i.e. climate feedback surrounded with large uncertainties), methane emissions due to its GHG and ozone precursor dual role, and the contribution from the stratosphere on tropospheric ozone under the RCP8.5 emission scenario. Moreover, here we use a synthetic ozone tracer to unambiguously identify for the first
time stratospheric- and tropospheric-produced ozone forcing. The CESM1-WACCM model, sensitivity simulations and ozone radiative effect calculations are described in Section 2. Results of future projected ozone changes and associated radiative effects are presented in Sect. 3 and 4, respectively. Different sources of uncertainties are discussed and accounted for in Sect. 5. Finally, a summary and concluding remarks
are presented in Sect. 6.



## 2    Methodology

### 2.1    Model description

We use the CESM version 1.1.1 with a configuration that fully couples the atmosphere and land components. A comprehensive description of the model is given by Marsh et al. (2013, and references therein).

The atmosphere component is WACCM version 4, a high-top model that extends from the surface to approximately 140 km in the lower thermosphere, with a vertical resolution ranging from 1.2 km near the tropopause to ~ 2 km near the stratopause, and horizontal resolution of 1.9º x 2.5º (latitude by longitude). The chemical scheme is the Model for Ozone and Related Chemical Tracers (MOZART) for the troposphere (Emmons et al., 2010) and the stratosphere (Kinnison et al., 2007), including recent updates (Lamarque et al., 2012; Tilmes et al., 2015). It includes 169 chemical species with detailed photolysis, gas-phase and heterogeneous reactions (see Tables A1 and A2 in Tilmes et al., 2016). Recent updates in the orographic gravity wave forcing – reducing the cold bias in Antarctic polar temperatures – (Calvo et al., 2017; Garcia et al., 2017) and the polar stratospheric chemistry (Wegner et al., 2013; Solomon et al., 2015) are included in the model. Concentrations of radiatively active gas-phase compounds such as ozone, nitrous oxide ($N_2O$), methane ($CH_4$) and other ODSs, are coupled to the model radiation scheme. Lightning-induced $NO_x$ ($LNO_x$) emissions are parameterized using the cloud top height method (Price and Vaughan, 1993), and annual global mean $LNO_x$ emissions are scaled to simulate present-day values of between 3-5 Tg N yr$^{-1}$.

A stratospheric ozone tracer (O3S) is implemented to represent the abundance and distribution of stratospheric-produced ozone in the troposphere (Roelofs and Lelieveld, 1997). O3S is equivalent to ozone in the stratosphere. In the troposphere it undergoes the same chemical loss processes as ozone, but does not undergo dry deposition, following the recommendations for the Chemistry-Climate Model Initiative (CCMI) (Eyring et al., 2013; Morgenstern et al., 2017). We apply an annual global correction factor to account for the dry deposition of O3S based on an additional model simulation. This correction factor is approximately linear, ranging from 0.7 at the surface to 0.95 around 250 hPa.



The land component is the Community Land Model version 4 which has the same horizontal resolution as the atmosphere component, and interactively calculates dry deposition for trace gases in the atmosphere (Val Martin et al., 2014) and biogenic emissions using the Model of Emissions of Gases and Aerosols from Nature
(MEGAN) version 2.1 (Guenther et al., 2012).

## 2.2   Experimental setup

This study uses time slice simulations driven by sea surface temperatures (SSTs) and sea ice climatologies from previous CESM1-WACCM fully coupled simulations (SENC2-8.5; see Morgenstern et al., 2017): 1990-2009 to represent the year 2000, and
2080-2099 to represent the end of the 21st century (nominally 2100). Each time slice experiment is integrated for 20 years, using the last 10 years in this study (i.e. the spin-up period covered the first 10 years). Seasonally varying boundary conditions are specified for carbon dioxide ($CO_2$), $N_2O$, $CH_4$, and ODSs (halogen-containing compounds), and the MACCity data set (Granier et al., 2011) is used to specify
anthropogenic $NO_x$, CO, and NMVOC emissions, as recommended for CCMI (Eyring et al., 2013). Volcanic eruptions are not included in the experiments, and the incoming solar radiation is fixed at 1361 $Wm^{-2}$. The quasi-biennial oscillation is imposed by relaxation of equatorial winds (90-3 hPa) with an approximate 28-month period between eastward and westward phase (Marsh et al., 2013).

Table 1 lists the simulations used in this study. The control simulation (CNTRL) had all boundary conditions set to the year 2000, and a set of sensitivity simulations had one (or more) boundary condition(s) changed to the year 2100. The simulations can be classified into three main groups:

1.  Sensitivity simulations that explore the impacts of climate change. Here SSTs,
sea ice and main GHGs (i.e. $CO_2$ and $N_2O$) are specified to year 2100 levels. The upper end emission scenario of the RCPs family is explored (RCP8.5). Natural biogenic emissions (e.g. isoprene) are calculated online, which are mainly governed by changes in $CO_2$, climate and land use (Squire et al., 2014). The indirect ozone radiative effect resulting from this climate feedback
is implicitly contained in the climate signal. However, unlike $LNO_x$ emissions it mainly impacts ozone in the lower troposphere, where ozone shows relatively small radiative efficiency (Rap et al., 2015). To isolate the impacts



of lightning-produced ozone, additional experiments are performed with year 2000 levels for LNO$_x$ emissions (fLNOx). Fixed LNO$_x$ simulations follow the Banerjee et al. (2014) approach, imposing the monthly mean LNO$_x$ emissions climatology from the CNTRL run and switching off its interactive calculation in the model. To justify this method, we compared temperature and tropospheric ozone fields between the CNTRL and CNTRL-fLNOx simulations and found negligible differences (not shown).

2. Stratospheric ozone recovery due to the slow decrease of ODSs (referring to the total organic chlorine and bromine species) concentrations regulated under the framework of the Montreal Protocol is investigated. Based on the CCMI recommendations, halogen species (CFC11, CFC12, CFC113, CFC114, CFC115, CCl4, HCFC22, HCFC141b, HCFC142b, CF$_2$ClBr, CF$_3$Br, CH$_3$Br, CH$_3$CCl$_3$, CH$_3$Cl, H1202, H2402, CH$_2$Br$_2$, and CHBr$_3$) are specified to year 2100 levels for the halogen scenario A1 (WMO, 2011), which includes the early phase-out of hydrochlorofluorocarbons agreed in 2007. Note that two brominated short-lived species (CH$_2$Br$_2$ and CHBr$_3$) were included in these experiments to accurately represent bromine loading and thus the associated ozone depletion, providing an additional bromine surface mixing ratio of ~ 6 pptv on top of that from the longer-lived bromine compounds.

3. Future levels of methane and its impacts on ozone are investigated. Concentrations of CH$_4$ are imposed to year 2100 levels from the RCP8.5 pathway – i.e. approximately double concentrations compared to year 2000. Note that methane levels were kept at year 2000 levels for the sensitivity simulations described above that explore climate change impacts.

## 2.3 Radiative effect

To calculate the RE of ozone we use the ozone radiative kernel of Rap et al. (2015). The radiative kernel is defined as the derivative of the radiative flux relative to perturbations in ozone. It is calculated by computing stratospherically adjusted ozone REs at the tropopause as a consequence of small perturbations (1 ppbv) imposed to each model layer (everything else is unchanged), with respect to the reference climatology. Multiplying simulated ozone fields with the ozone radiative kernel was





shown to be an efficient and accurate method to estimate ozone REs (Rap et al., 2015).

## 3    Present-day ozone radiative effects and model validation

A detailed present-day ozone evaluation of a similar model and experimental set-up was presented by Tilmes et al. (2016). In summary, simulated monthly mean ozone shows good agreement with observational estimates within 25 % range. Zonal and annual mean tropospheric ozone shows the best agreement with observations at low and mid-latitudes (±5 DU), a key region for its radiative effect (e.g. Rap et al., 2015).

Likewise, zonal and annual mean stratospheric ozone agrees fairly well with satellite estimates in the Southern Hemisphere (SH) and low latitudes (±30 DU), but larger deviations are found at mid- and high latitudes in the Northern Hemisphere (NH), a discrepancy also apparent in the models of the Atmospheric Chemistry and Climate Model Intercomparison Project (ACCMIP) (Iglesias-Suarez et al., 2016). The

tropospheric ozone budget (production, loss, dry deposition, stratospheric input), burden and lifetime for the CNTRL simulation (see Table 2 and Fig. S1) are within previous multi-model activities estimates (Stevenson et al., 2013; Young et al., 2013; Young et al., 2017).

         Figures 1a-1b show annual mean ozone RE calculated for the CNTRL

simulation (year 2000 or "present-day" hereafter) and the Tropospheric Emission Spectrometer (TES) from July 2005 until June 2008 (05–08). TES is the first product providing ozone profiles suitable for RE studies and has been previously evaluated against other observational estimates (e.g. Osterman et al., 2008), showing small bias in the troposphere and the stratosphere of approximately 3–4 DU. The annual and

global ozone RE in the CNTRL simulation is $1.38 \pm 0.1$ Wm$^{-2}$ (1 standard deviation associated with interannual variability unless otherwise specified), within the TES range of 1.35–1.41 Wm$^{-2}$. The spatial distribution of simulated and observed ozone REs are well correlated ($r = 0.7$, $p < 0.01$), although note that the noisier TES signal is largely the result of averaging only three years. Both the simulated and observed

present-day ozone REs reveal a positive poleward gradient, with a minimum in tropical regions (approximately 20º N/S) that is associated with the relatively low ozone levels found in the upper troposphere and lower stratosphere (see Fig. S1). A



peak is found at high latitudes in the NH, driven by transport of relatively rich tropospheric ozone air from mid-latitudes coupled with only moderate ozone depletion in the NH stratosphere. This is in contrast with a lower RE values within the SH polar vortex, driven by the larger stratospheric ozone depletion over Antarctica

(Solomon et al., 2015). Figure 1c compares the CNTRL annual mean ozone RE against the TES data set. Compared to TES, the simulated annual mean tends to overestimate the RE in the NH and underestimate it in the SH, consistent with the bias in the ozone distribution (Tilmes et al., 2016). Significant biases are mainly constrained to the tropical and subtropical regions – i.e. bias is defined here when the

simulated RE ±1.96 standard error (~95 % confidence interval) is outside the observed range. Tropical and subtropical regions are of particular interest for future changes in ozone and its resulting radiative forcing. Nevertheless, present-day ozone REs in these regions (30º N/S) are relatively small, with a large NH/SH compensation, as shown by the annual and global mean forcings.

## 4    Results

### 4.1    Ozone changes

Figure 2 shows annual and zonal mean ozone changes by 2100 compared to present-
day (CNTRL) imposing one single perturbation at a time. The discussion below guides through these changes, presenting the results from adding each successive perturbation, and includes chemistry-climate feedbacks.

The CLIMATE simulation (Fig. 2a) shows the expected pattern of ozone response (e.g. Kawase et al., 2011; Banerjee et al., 2014). In the troposphere, ozone
decreases primarily as a consequence of a warmer and more moist climate, which drives increased ozone loss via an enhanced $O(^1D)$ + $H_2O$ flux (Johnson et al., 2001). Reduced   net chemistry production is partially offset by an increase in the STE (Table 2), driven by an enhanced BDC (Zeng and Pyle, 2003). The fingerprint of this enhanced BDC can be seen in the lower stratosphere, both for decreases in the tropics
and increases at mid-latitudes, respectively associated with the enhanced ascending and descending regions (Hegglin and Shepherd, 2009). In this simulation, the 70 hPa



tropical (20° N/S) and zonal mean upwelling (Andrews et al., 1987) increases by 3.4 % dec$^{-1}$ compared to CNTRL (100 year trend). This trend is in agreement with current climate models projections of ~3.2 ± 0.7 % dec$^{-1}$ between 2005–2099 following the RCP8.5 (Hardiman et al., 2014). Additional ozone depletion over the Antarctic in CLIMATE compared to CNTRL is consistent with stratospheric cooling due to enhanced GHG levels (Fig. S3a), driving enhanced heterogeneous ozone loss chemistry (WMO, 2014). In contrast, cooling in the upper stratosphere results in ozone increases associated with a slowdown of catalytic $O_x$ cycles (Haigh and Pyle, 1982; Rosenfield et al., 2002).

Future $LNO_x$ emissions (+LIGHTNING; Fig. 2b) increase by ~33 %, which results in ozone increases mainly in the tropical and subtropical upper troposphere. However, present-day $LNO_x$ emissions have significant uncertainties and climate models do not agree even on the sign of the change due to different lightning parameterizations (Finney et al., 2016). Nevertheless, simulated present-day $LNO_x$ emissions of 4.8 ± 1.6 Tg(N) yr$^{-1}$ lies within observationally-derived estimates, and the model's $LNO_x$ sensitivity to climate of 10.8 % K$^{-1}$ is at the upper end of the two standard deviation climate model range (8.8 ± 2 % K$^{-1}$) (Finney et al., 2016). The net global tropospheric ozone responses to climate will be largely determined by the interplay between climate-induced ozone losses and lightning-induced ozone production.

Reductions in inorganic chlorine and bromine abundances (++O3-RECOVERY; Fig. 2c) result in stratospheric ozone increases. Upper stratospheric ozone recovers largely due to decreases in $ClO_x$-catalysed ozone destruction. Due to reduced heterogeneous ozone loss chemistry, the largest changes are found in polar regions in the lower stratosphere, with increases of ~450 % over the Antarctic (November) and ~45 % over the Arctic (April). Greater abundances of stratospheric ozone result in an approximately 20 % increase in the STE (Table 2) driving higher levels of tropospheric ozone, particularly at mid- and high latitudes in the SH (related to ozone hole recovery) and tropical and subtropical upper troposphere (the descending region of the BDC), which is consistent with previous model estimates (Banerjee et al., 2016). The BDC-driven increases are somewhat offset by the larger overhead ozone column reducing actinic fluxes and therefore ozone photochemical production (Table 2) (Banerjee et al., 2016).



Methane is a greenhouse gas, an ozone precursor in the troposphere and plays various roles in the stratosphere, and these processes are difficult to isolate from the rest. The +++METHANE simulation (Fig. 2d) shows a widespread increase of ozone in the troposphere, with annual and global tropospheric column ozone increase of 15 ± 8 % (Table S1). Previous modelling studies reported similar increases of 10–13 % (Brasseur et al., 2006; Kawase et al., 2011). Compensation between ozone decreases in the upper stratosphere (enhanced $HO_x$-catalysed chemistry) and increases in the lower stratosphere (smog-like chemistry and the partitioning of active/inactive chlorine) (Randeniya et al., 2002; Stenke and Grewe, 2005; Portmann and Solomon, 2007; Fleming et al., 2011; Revell et al., 2012), results in small changes of 2 ± 5 % for the annual and global stratospheric column ozone.

### 4.2  Ozone radiative forcing

Figure 3 shows maps of annual mean radiative forcing between 2000 and 2100 due to changes in ozone for the whole atmosphere, along with zonal mean forcings associated with changes in the troposphere and the stratosphere for single perturbation simulations. Note that zonal mean forcings are latitudinally-weighted, allowing direct comparison with the total forcing. Annual and global mean forcing values and their standard deviation (i.e. due to ozone changes only) are listed in Table 3. Ozone radiative forcing shows strong dependence on the vertical distribution of the change (e.g. Lacis et al., 1990; Forster and Shine, 1997; Rap et al., 2015) and to a lesser extent on the horizontal distribution (e.g. Berntsen et al., 1997). Differences can be seen in both the geographical pattern of the forcing and in the magnitude related to the drivers.

The global forcing associated with CLIMATE (Fig. 3a) of 41.3 ± 75 mWm$^{-2}$ is small and not significant. The geographical pattern shows a relatively strong and significant forcing at high latitudes in the NH, related to ozone increases in the lower stratosphere (transport from enhanced BDC) and upper stratosphere (reduced chemical loss due to cooling). However, this is outweighed by a negative tropospheric forcing in the tropics and a negative stratospheric forcing in the SH extra-tropical region. The latter is largely due to additional ozone depletion in the lower stratosphere (i.e. reduction of STE; not shown).



Future lightning-induced $NO_x$ emissions (+LIGHTNING; Fig. 3b) shows relatively large and significant global ozone forcing of $104.2 \pm 79$ mWm$^{-2}$, mainly the result of simulated tropospheric ozone changes of $2.1 \pm 2.3$ DU. Two distinct peak regions are evident around the subtropical belts, where large ozone changes are

coincident with relatively cloud-free areas, higher temperature, and a low solar zenith angle. The strongest positive forcing is found over the Sahara and Middle East deserts, associated with greater surface albedo.

Ozone recovery (++O3-RECOVERY) drives a significant forcing of $129.2 \pm 81$ mWm$^{-2}$ (Table 3). This forcing is largely confined to the mid- and high

latitudes, particularly in the SH (due to ozone hole recovery), and is mainly linked to the stratosphere (Fig. 3c). Extra-tropical STE is especially important in the SH. This is demonstrated by tropospheric forcing of about $\sim$100 mWm$^{-2}$ in this region, which is largely the result of stratospheric-produced ozone transported to the troposphere.

The +++METHANE simulation shows a large positive forcing around the

subtropical belts (Fig. 3d), which is principally confined to the troposphere, as there is a compensation between changes in the lower and upper stratosphere (Fig. 2d). In the tropical and subtropical troposphere, methane is more readily oxidised due to larger OH levels, which results in relatively large ozone increases (Fig. 3d). In addition, significant forcings at high latitudes, particularly over the Arctic, are linked to the

stratosphere (i.e. reduced ozone loss via decreased active/inactive chlorine partitioning).

Figure 4 shows maps of annual mean normalised tropospheric ozone radiative forcing (NRF) between 2000 and 2100 for the four sensitivity simulations. The NRF – defined here as the tropospheric ozone radiative forcing divided by the tropospheric

column ozone – is a useful diagnostic to gain insight into radiative effects of ozone changes. Very similar global NRFs of $\sim$39 mWm$^{-2}$DU$^{-1}$ due to climate and methane, indicates relatively evenly distributed ozone changes in the troposphere. In contrast, more localised lightning-produced ozone results in higher global NRF of 46 mWm$^{-2}$ DU$^{-1}$, whereas ozone increases at high latitudes due to ozone recovery

results in smaller NRF of 35 mWm$^{-2}$ DU$^{-1}$. This highlights the dependence of the resulting forcings on the vertical and horizontal distribution changes of ozone.





Previous studies have shown that the radiative forcing from tropospheric and stratospheric ozone do not have distinct drivers (Søvde et al., 2011; Shindell et al., 2013). Our results support this and show that climate change, ODSs and methane have consequences for both tropospheric and stratospheric ozone radiative forcing (Table 3). In this set of simulations, changes in ozone occurring in the troposphere and the stratosphere respectively contribute ~75 % and 25 % to the total annual and global forcing of $417.6 \pm 81$ mWm$^{-2}$.

Further insight can be gained by attributing ozone forcing based on its origin in the stratosphere or the troposphere. In these simulations, we used a stratospheric ozone tracer (see Sect. 2) to unambiguously differentiate ozone with tropospheric origin (O3T) from that with stratospheric origin (O3S). Table 3 shows such "source classified" ozone radiative forcings, using the "O3S/ozone" and "O3T/ozone" ratios for tropospheric and stratospheric forcings respectively. Stratospheric-produced ozone contributes to ~47 % of the annual and global future ozone forcing in this set of simulations, which strongly reinforces the importance of stratospheric-tropospheric interactions.

### 4.3 Methane feedback and resulting ozone forcing

Future climate change and emissions of ODSs and methane will affect the oxidising capacity of the atmosphere (e.g., via hydroxyl radicals, OH), which in turn influences the methane lifetime ($\tau$CH4) on decadal time scales resulting in a "long-term" response of tropospheric ozone (e.g. Fuglestvedt et al., 1999; Wild and Prather, 2000; Holmes et al., 2013). The simulations considered here neglect this feedback by imposing fixed and uniform lower boundary conditions for methane of 1748 and 3744 ppbv for years 2000 and 2100 respectively. However, we can estimate how methane concentrations would have adjusted if they were free to evolve, as well as the associated ozone response and radiative forcing. Using the method described by Fiore et al. (2009, and refences therein), we calculate equilibrium methane abundances, [CH4]eq, by

$$[CH_4]_{eq} = [CH_4]_{CNTRL} \times \left( \frac{\tau CH_4(p)}{\tau CH_4(r)} \right)^f \qquad (1)$$



where CNTRL represents the fixed boundary conditions for year 2000; *(p)* and *(r)* refer to the perturbation and reference simulations respectively; and $f$ is a feedback factor which accounts for the response of methane to its own lifetime. The feedback factor is explicitly calculated for WACCM using the ++O3-RECOVERY "(a)" and +++METHANE "(b)" simulations, as follows

$$f = \frac{1}{(1-s)} \tag{2}$$

where $s$ is calculated by

$$s = \frac{[\ \ln(\tau CH_4(b)\ ) - \ln(\tau CH_4(a)\ )]}{[\ln(BCH_4(b)\ ) - \ln(BCH_4(a)\ )]} \tag{3}$$

and where $BCH_4$ is the annual and global mean methane burden. We calculate a value of $f$ of 1.43 which is at the mid-upper end of the literature range (1.19–1.53) (Prather et al., 2001; Stevenson et al., 2013; Voulgarakis et al., 2013) and within 7 % of the observationally constrained best estimate of 1.34 (Holmes et al., 2013).

The ozone response to this methane feedback is estimated by linear interpolation:

$$\Delta O3(eq - CNTRL) = \left[\frac{\Delta CH_4(eq - CNTRL)}{\Delta CH_4(b-a)}\right] \times \Delta O3(b-a) \tag{4}$$

where $\Delta O3$ is the change in annual and global mean of tropospheric column ozone (Table S1). The associated tropospheric ozone forcings to methane feedback are given by the product of $\Delta O3$ and the NRF due to methane perturbation (39 mWm$^{-2}$ DU$^{-1}$; Fig. 4d) and are shown in Table 3. The overall long-term tropospheric ozone forcing related to the methane feedback in this set of simulations is a moderate increase of ~15 %. Climate change (CLIMATE and +LIGHTNING simulations) enhances the oxidising capacity of the atmosphere, which results in a small negative forcing of −19 mWm$^{-2}$. In the +++METHANE simulation, OH concentrations are strongly reduced and the associated forcing of 63 mWm$^{-2}$ outweighs the climate forcing. This forcing is within the range of ~76 (40-146) mWm$^{-2}$ of the ACCMIP ensemble (Stevenson et al., 2013), when considering the same change in methane concentrations.





### 4.4 Background conditions and forcing

Since the ozone response to a given perturbation is dependent on the background conditions (e.g. temperature, radiative heating, trace gas levels), the resulting forcing associated to individual drivers may be sensitive to the experimental design. For example, lightning-induced ozone forcing due to climate change may differ significantly under present-day or doubled methane concentrations (i.e. year 2000 or year 2100-RCP8.5 abundances). In the present study, we imposed single perturbations successively. Therefore, the total ozone forcing calculated from this set of simulations includes chemistry-climate feedbacks arising from the interactions between the various perturbations. Yet the attribution of indirect ozone forcings to individual drivers may be sensitive to the order considered (Table 1).

We also completed an additional set of simulations (Table S2) to assess the robustness of the calculated RF to the order the perturbations were applied (Table 3). Lightning-induced net ozone forcing (+LIGHTNING; $104.2 \pm 79$ mWm$^{-2}$) is not significantly different at the 95 % confidence interval (due to interannual variability only unless otherwise specified) compared to that calculated under approximately doubled methane concentrations (CLIMATE[CH4-2100] minus CLIMATE[CH4-2100]-fLNOx). Although the +LIGHTNING net ozone forcing is 50 mWm$^{-2}$ lower relative to the latter, both lie within the interannual uncertainty (~80 mWm$^{-2}$). The above forcing associated with ozone recovery (++O3-RECOVERY; $129.2 \pm 81$ mWm$^{-2}$) is calculated under climate change (i.e. including lightning feedbacks) and present-day methane concentrations, though it also can be derived under present-day climate (ODS minus CNTRL) or doubled methane concentrations (+++METHANE minus CLIMATE[CH4-2100]). We find no significant differences between the forcings associated with these background conditions, although the mean forcing resulting from the ++O3-RECOVERY is greater by ~20 mWm$^{-2}$. Finally, methane-induced net ozone forcing due to doubling its concentrations relative to present-day under ozone recovery conditions (+++METHANE; $225.5 \pm 85$ mWm$^{-2}$), is not significantly different to that under present-day ODSs concentrations (CLIMATE[CH4-2100] minus +LIGHTNING) or without lightning feedbacks (CLIMATE[CH4-2100]-fLNOx minus CLIMATE). The forcing associated with +++METHANE lies within the latter forcings (i.e. 50 mWm$^{-2}$ range). Therefore, we conclude that future ozone forcings due to lightning, ozone recovery and methane



concentrations – presented in Table 3 – are robust, with regard to background conditions.

The fact that global and annual ozone forcings associated with single perturbations are not significantly different with regard to background conditions is perhaps somewhat surprising, given that, for instance, ozone production is sensitive to the relative abundances of volatile organic compounds and $NO_x$ (e.g. Sillman, 1999). However, while the globally averaged forcing is not significantly affected by the order in which the perturbations are considered, there are significant differences in budget terms (e.g. ozone burden differences due to lightning can be as large as $4.5 \pm 1.4$ Tg), as well as ozone levels in particular regions of the atmosphere. Therefore, the non-linear additivity of the perturbations is important when considering their impacts on quantities such as ozone profiles and surface air quality (not shown).

## 5   Uncertainties and outlook

We calculate a net ozone radiative forcing of $417.6 \pm 81$ mWm$^{-2}$ corresponding to the year 2100 compared to present-day, with the one standard deviation uncertainty arising from variability in ozone between the years of the time slice simulations. This variability indicates a $\pm 19$ % uncertainty, which is similar to the spread across the ACCMIP ensemble (Stevenson et al., 2013). However, different sources of uncertainty exist in the ozone forcing. Background conditions affect somewhat the resulting ozone forcings (see Sect. 4.4), and therefore add an extra $\pm 2.5$ % to the overall uncertainty. Previously, uncertainties arising from the tropopause definition ($\pm 3$ %), the radiation scheme or forcing calculation ($\pm 10$ %), and the extent to which clouds and stratospheric temperature adjustment influence ozone forcing ($\pm 7$ % and $\pm 3$ % respectively) have been estimated (Stevenson et al., 2013). Climate feedbacks, land-use change, natural ozone precursor emissions, and future changes in the structure of the tropopause (Wilcox et al., 2012) may introduce at least an additional $\pm 20$ % uncertainty (Stevenson et al., 2013). Following Stevenson et al. (2013), we assume that the above individual uncertainties are independent and combine them to estimate an overall uncertainty of $\pm 30$ %, which represents the 95 % confidence interval. Note that Skeie et al. (2011) from an independent analysis estimated the same overall uncertainty.



Figure 5 summarises the global and annual net ozone forcing as well as the forcings by driver and region. Overall, our annual global mean best estimate for the net ozone radiative forcing between 2000 and 2100 is $420 \pm 130$ mWm$^{-2}$, with tropospheric and stratospheric forcings of $300 \pm 90$ mWm$^{-2}$ and $120 \pm 40$ mWm$^{-2}$, respectively. Current estimates for tropospheric and stratospheric ozone forcings from 1750 to 2011 are $400 \pm 20$ mWm$^{-2}$ and $-50 \pm 100$ mWm$^{-2}$, respectively (Myhre et al., 2013). An increase of 0.5 DU in tropospheric ozone was estimated in Skeie et al. (2011) from 2000 to 2010, and a tropospheric ozone normalized radiative forcing of 42 mWm$^{-2}$ DU$^{-1}$ calculated from the ACCMIP ensemble (Stevenson et al., 2013). Therefore, we estimate a net ozone forcing of $750 \pm 230$ mWm$^{-2}$ from 1750 to 2100 based on our simulations, which is the result of the forcings in the troposphere and the stratosphere ($690 \pm 210$ mWm$^{-2}$ and $60 \pm 20$ mWm$^{-2}$ respectively). Our tropospheric forcing is within the range estimated from the ACCMIP models of $620 \pm 190$ mWm$^{-2}$ (Stevenson et al., 2013).

Previous work has shown that NRF is an appropriate tool for estimating annual and global tropospheric forcings derived from changes in tropospheric column ozone, which in turn reduces the multi-model uncertainty (Gauss et al., 2003). The NRF in our analysis of 43 mWm$^{-2}$ DU$^{-1}$ is similar to that from the ACCMIP models between the 1850s and 2000s, but larger compared to that in Gauss et al. (2003). This supports the future tropospheric ozone forcings and their uncertainties during the 21st century derived from the ACCMIP ensemble (calculated using the NRF), and may be used as a benchmark for individual studies. The net ozone NRF is 18 mWm$^{-2}$ DU$^{-1}$ (including changes in the stratosphere). The smaller net ozone NRF compared to the tropospheric ozone NRF is mainly caused by the fact that ozone changes in the upper stratosphere have a relatively negligible impact on RF. While this is in agreement with a previous multi-model study (Gauss et al., 2003), their larger value of 29 mWm$^{-2}$ DU$^{-1}$ is likely associated with their focus on the lower stratosphere only, where ozone is more radiatively efficient than in the upper stratosphere.

Although previous studies have examined key drivers of ozone during the 21st century and future changes are relatively well understood (e.g. Kawase et al., 2011; Banerjee et al., 2014; Banerjee et al., 2016), the resulting forcings have been explored in less detail (e.g. Gauss et al., 2003; Stevenson et al., 2013). Following a process-based approach that includes chemistry-climate feedbacks, we calculate that climate-





only, lightning, ozone recovery and methane emissions contribute respectively −10 ± 18 %, 25 ± 19 %, 31 ± 19 %, and 54 ± 20 % to the net ozone RF between 2000 and 2100 (Table 3 and Fig. 4). Further uncertainties arise from the long-term ozone response to methane changes, which could increase the overall tropospheric forcing by ~15 %. Climate change (including lightning feedbacks) leaves a small tropospheric ozone forcing of 64.1 ± 44 mWm$^{-2}$. A subset of eight models from the ACCMIP activity shows a small negative but not significant tropospheric forcing of −33 ± 42 mWm$^{-2}$, with few models reporting positive forcings (Stevenson et al., 2013). The impact of climate change on ozone forcing is surrounded by large uncertainties, which may be associated with chemistry-climate feedbacks and the lack of confidence in the LNO$_x$ sensitivity to global mean surface temperature, due to different parameterizations and the vertical distributions of the emissions (Banerjee et al., 2014; Finney et al., 2016), as well as changes in the BDC (Butchart, 2014). For example, the climate change-induced net ozone forcing between 2000–2100 – following the future emission scenario RCP8.5 in an independent CCM – is of the same order of magnitude but different sign (−70 mWm$^{-2}$) (Banerjee et al., 2017). While they found similar tropospheric ozone forcing of 80 mWm$^{-2}$, their negative stratospheric ozone forcing outweighs the latter (−150 mWm$^{-2}$). Methane- and ODSs-induced ozone forcings have respectively a substantial contribution from the stratosphere (~14 %) and the troposphere (~34 %), recently shown in modelling studies (Søvde et al., 2011; Shindell et al., 2013; Banerjee et al., 2017). A striking result, however, is the contribution of the stratospheric-produced ozone to the net forcing of ~30 ± 17 % and ~99 ± 59 % due to methane and ODSs emissions, respectively. This reflects the roles that methane plays in stratospheric ozone chemistry (i.e. particularly in the lower stratosphere), and that ozone recovery principally occurs in the stratosphere.

## 6 Summary and conclusions

This study has explored future changes in ozone by the end of the 21st century and the resulting radiative forcing following a process-based approach, imposing one forcing at a time. We have used the RCP8.5 to represent an upper limit on these responses. This is a different approach to previous studies, which typically have either explored future changes in ozone concentrations or ozone forcing. The methane feedbacks (due



to the changing oxidising capacity of the atmosphere, and due to the long-term tropospheric ozone response) and its forcing have also been accounted for. In addition, non-linearities arising from chemistry-climate interactions have been investigated.

The simulated present-day ozone radiative effect is in good agreement with estimates based on observed ozone from TES, particularly in terms of its spatial distribution. However, there are systematic biases: RE is overestimated in the NH and underestimated in the SH, with significant biases in the subtropics. These RE biases are mostly consistent with the biases in tropospheric ozone in current climate models

(Young et al., 2017), although the simulated annual global present-day tropospheric column ozone (28.9 ± 1.5 DU) is within observed interannual variability of 28.1–34.1 DU (Young et al., 2013). The fact that the biases are apparent in many climate models suggests a common deficiency, and emissions data have been proposed as a likely candidate (Young et al., 2013; Young et al., 2017).

Our analysis shows that the ozone radiative forcing arising from climate driven changes is small and not significant ($63 \pm 76$ mWm$^{-2}$), which is largely the result of the interplay between lightning-produced ozone and enhanced ozone destruction (via increased temperatures and humidity). Higher methane concentrations and reduced ODSs levels also have consequences for ozone forcing in the stratosphere

($32 \pm 31$ mWm$^{-2}$) and the troposphere ($46 \pm 47$ mWm$^{-2}$) respectively. We have demonstrated both the importance of stratospheric-tropospheric interactions and the stratosphere as a key region controlling a large fraction of the tropospheric ozone forcing (i.e. from the source point of view compared to the more common division by recipient-region).

Future annual and global tropospheric and stratospheric column ozone changes from year 2000 to 2100 in this set of simulations (7.0 DU and 21.3 DU respectively) are mainly driven by methane and ODSs emissions, respectively (Table S1). These changes lead to a net ozone radiative forcing of $420 \pm 130$ mWm$^{-2}$ compared to present-day, with an overall uncertainty of ±30 % (i.e. representing the

95 % confidence interval). Relative to the pre-industrial period (year 1750), our best estimate for the year 2100 net ozone radiative forcing is $750 \pm 230$ mWm$^{-2}$.



This study highlights the key role of the stratosphere in determining future ozone radiative forcing in spite of the fact that the impacts largely take place in the troposphere. Increasing confidence in present-day observations of the Brewer-Dobson circulation and the stratospheric-tropospheric exchange will therefore play a crucial role in improving climate models and better constraining ozone radiative forcing. A future study will address the importance of the stratosphere on future air quality commitments, which may better inform emission regulations.

*Acknowledgements*. This work was supported by NERC, under project number NE/L501736/1. F. Iglesias-Suarez would like to acknowledge NERC for a PhD studentship and thank F. Govantes for hosting him at the Centro Andaluz de Biología del Desarrollo (CABD) while he completed some of this work. WACCM is a component of the Community Earth System Model (CESM), which is supported by the NSF and the Office of Science of the U.S. Department of Energy. Computing resources were provided by NCAR's Climate Simulation Laboratory, sponsored by the NSF and other agencies. This research was enabled by the computational and storage resources of NCAR's Computational and Information Systems Laboratory (CISL). We thank the NASA JPL TES team for releasing the TES ozone.



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



Table 1. Summary of the model simulations

| Simulation | Climate[1] | ODSs[2] | CH$_4$[3] |
|---|---|---|---|
| CNTRL | 2000 | 2000 | 2000 |
| CLIMATE | 2100 (fLNOx)[4] | 2000 | 2000 |
| +LIGHTNING | 2100 | 2000 | 2000 |
| ++O3-RECOVERY | 2100 | 2100 | 2000 |
| +++METHANE | 2100 | 2100 | 2100 |
| CNTRL+fLNOx | 2000 (fLNOx)[4] | 2000 | 2000 |

[1] Climate (SSTs, sea ice, CO$_2$ and N$_2$O, if not otherwise specified).

[2] Relative to CNTRL, ++O3-RECOVERY simulation is driven by ODSs boundary conditions of −63.2% (2.156 ppbv) total chlorine, −35.7% (8.1 pptv) total bromine and −67.6% (1.376 ppbv) total fluorine.

[3] Relative to CNTRL, +++METHANE simulation is driven by CH$_4$ boundary conditions of 214.2 % (3744 ppbv).

[4] Offline lightning-induced NO$_x$ emissions imposed by applying a monthly mean climatology of the CNTRL simulation.





**Table 2.** Tropospheric ozone budget, including: Ozone production (P) and loss (L) terms are based on the gas-phase reaction rates of the $O_x$ family ($O_3$, O, $O_1$D and $NO_2$); Net chemical production of ozone is defined as the residual of the production and loss terms (N = P − L); Dry deposition of ozone (D) term; Stratospheric–Tropospheric exchange (S; i.e. influx from the stratosphere) term is the residual of the dry deposition and net chemistry production terms (S = D − N); Ozone burden (B) term; Ozone and methane lifetimes ($\tau_{O3}$, $\tau_{CH4}$ respectively) is the ratio between the burden and total losses ($\tau$ = burden / total loss); $\tau_{CH4}$ includes loss with respect to OH and adjusted for soil uptake (160 years) and stratospheric sink (120 years) (Prather et al., 2012); Burden for the stratospheric ozone tracer ($B_{O3S}$) after corrected to include dry deposition (not shown).

| Simulation | P (Tg yr$^{-1}$) | L (Tg yr$^{-1}$) | N (Tg yr$^{-1}$) | D (Tg yr$^{-1}$) | S (Tg yr$^{-1}$) | B (Tg) | $\tau_{O3}$ (years) | $\tau_{CH4}$ (years) | $B_{O3S}$ (Tg) |
|---|---|---|---|---|---|---|---|---|---|
| ACCENT (year 2000) | 5110 ± 606 | 4668 ± 727 | 442 ± 309 | 1003 ± 200 | 552 ± 168 | 344 ± 39 | 22.3 ± 2.0 | 8.7 ± 1.3 | ---- |
| ACCMIP (year 2000) | 4877 ± 853 | 4260 ± 645 | 618 ± 275 | 1094 ± 264 | 477 ± 97 | 337 ± 23 | 23.4 ± 2.2 | 8.5 ± 1.1 | ---- |
| CNTRL | 4678 | 4195 | 483 | 881 | 398 | 318 | 22.9 | 7.2 | 123 |
| CLIMATE | 5111 | 4809 | 302 | 811 | 510 | 309 | 20.1 | 6.9 | 119 |
| +LIGHTNING | 5378 | 5057 | 322 | 833 | 511 | 329 | 20.4 | 6.6 | 138 |
| ++O3–RECOVERY | 5303 | 5058 | 245 | 855 | 610 | 337 | 20.8 | 6.6 | 131 |
| +++METHANE | 6072 | 5759 | 313 | 979 | 666 | 378 | 20.5 | 8.3 | 138 |
| CNTRL+fLNOx | 4648 | 4169 | 479 | 878 | 399 | 313 | 22.7 | 7.2 | 121 |





Table 3. Global and annual mean ozone RF and the standard deviation[a] (mWm$^{-2}$) by driver and region, relative to CNTRL (year 2000).

| | Whole-atmosphere | Region | | Source | | CH$_4$[b] |
| --- | --- | --- | --- | --- | --- | --- |
| | | Tropo. | Strat. | Tropo. | Strat. | Tropo. |
| CLIMATE | −41.3 ± 75 | −40.4 ± 43 | −0.8 ± 27 | −19.8 ± 22 | −21.4 ± 48 | −7.7 |
| +LIGHTNING | 104.2 ± 79 | 104.5 ± 45 | −0.3 ± 29 | 79.4 ± 35 | 24.8 ± 39 | −11.3 |
| ++O3-RECOVERY | 129.2 ± 81 | 45.9 ± 47 | 83.2 ± 30 | 0.8 ± 0.7 | 128.3 ± 76 | 1.9 |
| +++METHANE | 225.5 ± 85 | 193.1 ± 51 | 32.4 ± 31 | 160.8 ± 43 | 64.7 ± 39 | 63.1 |
| Total | 417.6 ± 81 | 303.2 ± 48 | 114.6 ± 30 | 221.3 ± 19 | 196.3 ± 33 | 46.0 |

[a] The annual global mean is given along with the (±) standard deviation (i.e. associated with ozone variability).

[b] Long-term ozone forcing due to methane chemistry-climate feedback.




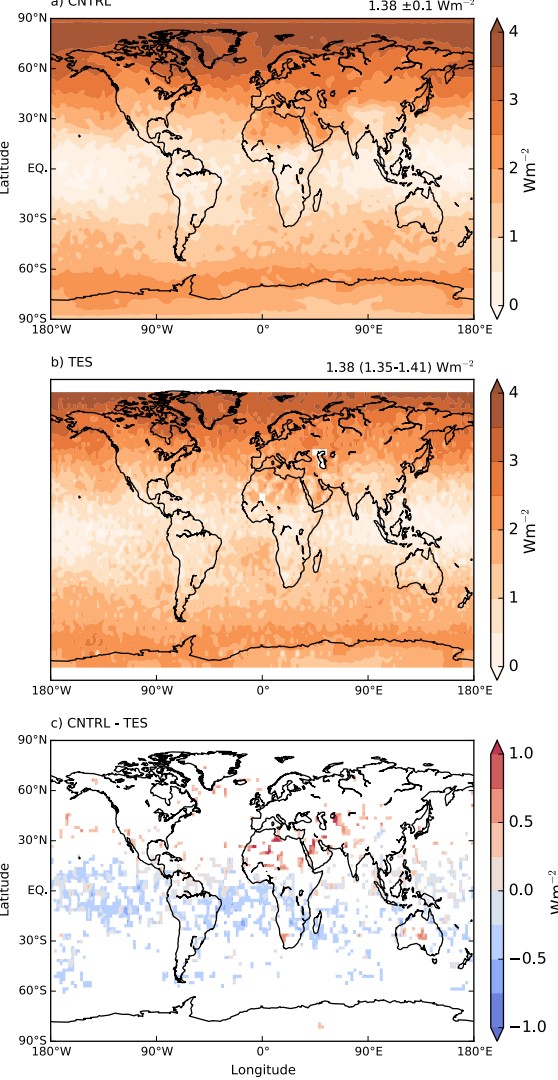

Figure 1. Comparison of the annual mean ozone radiative effect between (a) the CNTRL simulation and (b) the Tropospheric Emission Spectrometer (TES) from July 2005 until June 2008 (05–08). (c) CNTRL simulation bias compared to the TES. Differences are masked for the ±1.96 standard error within the three years observed range.



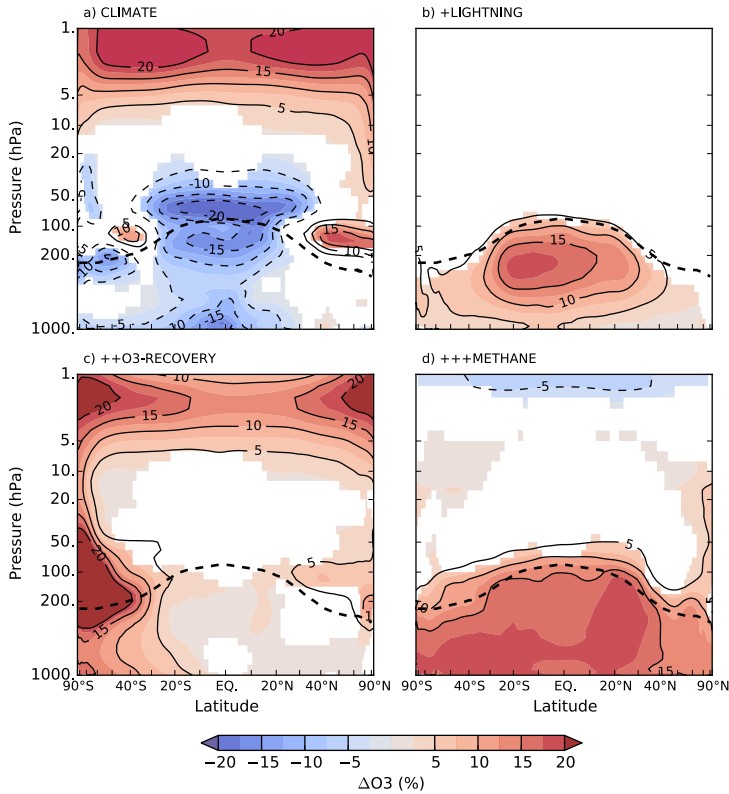

Figure 2. Changes in annual and zonal mean ozone due to (a) CLIMATE (b) +LIGHTNING, (c) ++O3-RECOVERY, and (d) +++METHANE by successively adding each individual perturbation to the CNTRL simulation. Contour colours are for statistically significant changes at the 95 % confidence interval using two-tailed Student's t test. The black solid line represents the chemical tropopause based on the CNTRL 150 pbb ozone contour.



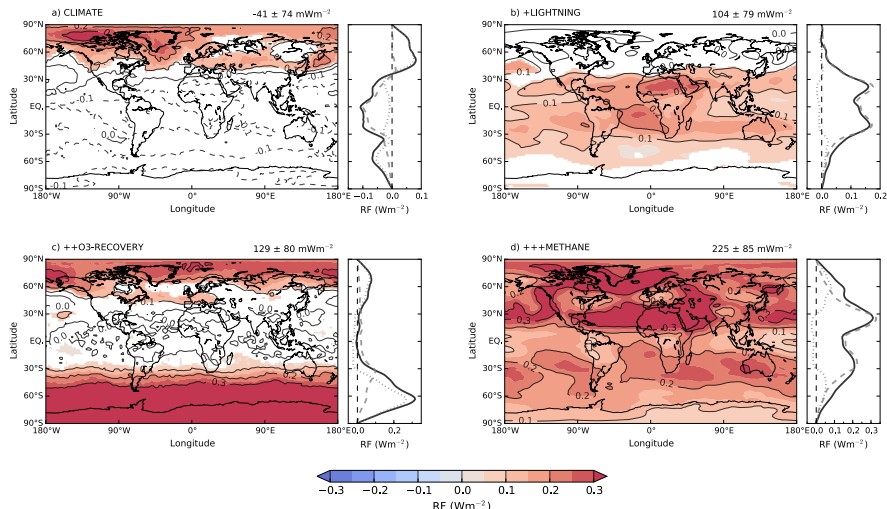

Figure 3. Annual mean maps of ozone radiative forcing (whole atmosphere) due to (a) CLIMATE (b) +LIGHTNING, (c) ++O3-RECOVERY, and (d) +++METHANE. Contour colours are for statistically significant changes at the 95 % confidence interval using two-tailed Student's t test. The annual and global mean is shown on the top right corner (mWm$^{-2}$). Right panels show zonal mean ozone forcings for the whole atmosphere (solid black), troposphere (dashed grey), and stratosphere (dotted grey). The zonal mean forcings are latitudinally-weighted, i.e. cosine(latitudes).





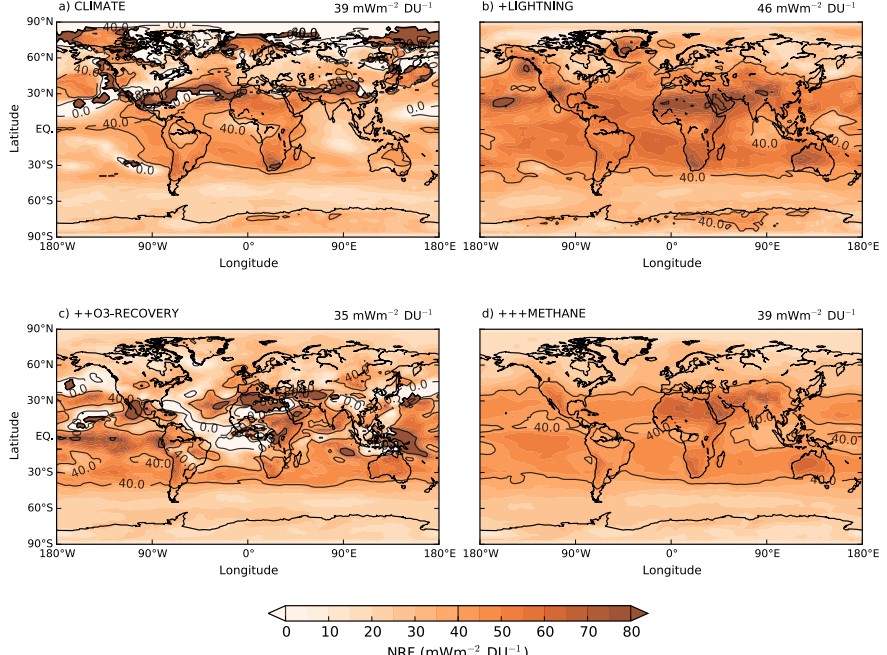

Figure 4. Annual mean maps of normalised tropospheric ozone radiative forcing (i.e. divided by the tropospheric column ozone change) due to (a) CLIMATE (b) +LIGHTNING, (c) ++O3-RECOVERY, and (d) +++METHANE. The annual and global mean is shown on the top right corner (mWm$^{-2}$).





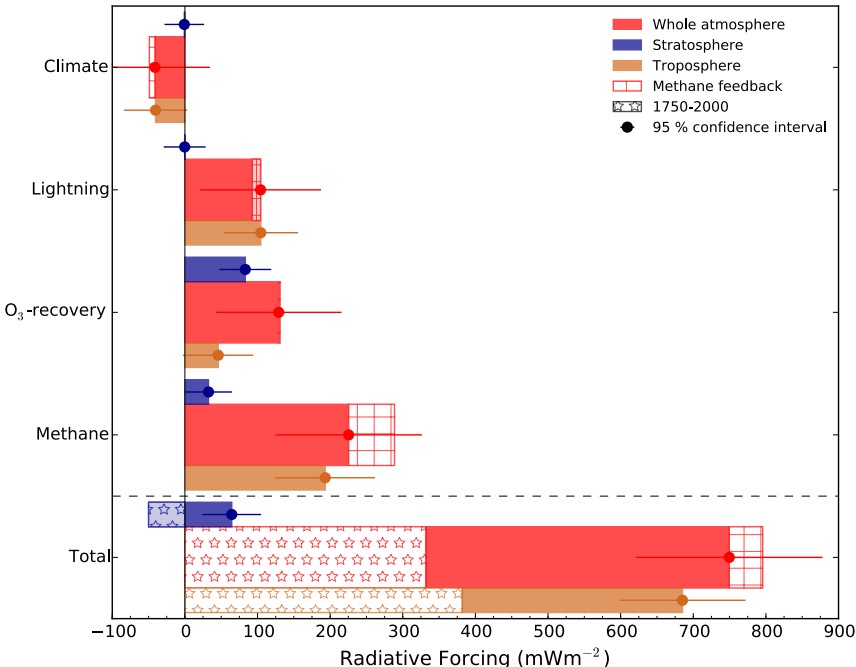

Figure 5. Ozone radiative forcings by drivers (2000–2100; mWm$^{-2}$). Tropospheric (brown), stratospheric (blue) and net (whole atmosphere, red) forcings are shown. Associated ozone forcings to methane feedback (square-hatched) are shown along with the net forcings. The overall ozone forcing (TOTAL) is also scaled to 1750 (1750–2000; star-hatched). Dots and error bars indicate the mean and the 95 % confidence intervals of the forcings respectively (sources of uncertainty are detailed in Sect. 5).