# Peer review of "Key drivers of ozone change and its radiative forcing over the 21st century"

_Atmospheric Chemistry and Physics, 2017_

## Referee Comment (RC1) · Anonymous Referee #1 · 22 Jan 2018

Review of "Key drivers of ozone change and its radiative forcing over the 21st century" byy F. Iglesias-Suarez et al.

General Comments

This is an interesting and generally well-constructed paper that presents new results and is well worthy of publication. However, several things about the paper confused me, and I think must be clarified before it is accepted for publication. These are listed in detail below, but can be broadly summarised as follows. The paper is about modelled ozone changes between 2000 and 2100, but after reading it I am unclear how the major influence of changing anthropogenic emissions is included (the focus is on changes in climate, lightning, ozone-depleting substances, and methane). I think they must change, based on some of the text, and comparisons to other studies, but I can't work

out how they are handled from what is written in the text. There are a few other things that I find a bit confusing, but I think if the authors can address the specific comments below the paper will be good for publication.

Specific Comments

P2 l4-6: Do changes in the stratosphere account for 47% of the overall RF for both 1750-2000 and 2000-2100?

P2 l15: 90% by mass.

P2 l19: It is difficult to agree that ozone is 'the main' concern for climate change and air quality ($CO_2$ and PM are probably more important!) – suggest change to 'an important'.

P2 l26: Delete comma after although.

P2 l30: "BDC is the wave-driven factor" – rephrase.

P3 l13: ODS (not ODSs).

P3 l18: troposphere (not stratosphere).

P3 l21-24: Clarify – do reductions in O3 photolysis mean more tropospheric O3?

P4 l1: to -> with.

P4 l4: lower -> reduced.

P4 l9: What about aircraft NOx?

P5 l15-17: Aren't higher latitudes than the tropics more strongly influenced by stratospheric influx of O3?

P5 l27-29: Sentence unclear – reword.

P6 l8: i.e. -> e.g. (also following lines 9 and 11).

P6 l19-23: Unclear – reword.

P7 l19: Is CH4 an ODS? (I don't think so.)

P7 l27-28: It may have been recommended by CCMI, but I don't really understand why the O3S tracer can't also be lost by O3 dry deposition (like any ozone molecule). Can you clarify why? This seems like an odd approach. I see you apply a correction factor – I guess to account for this.

P8 (This should be somewhere in the model description or experimental set-up): Is ozone coupled radiatively to the climate model? I assume it is, and this means that any changes in ozone generate changes in meteorology. This should be made clear, as it has important implications for how the results are interpreted.

P8 l10: There is a minor inconsistency in your naming/approach – 1990-2009 is used to represent 2000; 2080-2099 is used to represent 2100. Why not nominally 2090? Actually on the next line you say each experiment is 20 years with only the last 10 years used – so 2000-2009 or the 2000s and 2090-2099 i.e. the 2090s?

P8 l14: You use MACCity anthropogenic emissions, but don't make it clear if they are held fixed at year 2000 levels or if they change between 2000 and 2100 (RCP8.5). This needs to be clear as it is also very important.

P8 l19 phases

P8 l23: Clarify if land-use is changing in these runs – if so this would have implications for, e.g., BVOC emissions, etc.

P10 l4: I think Section 3 belongs within the "Results", rather than prior to it. It contains some results!

P10 l7: a 25%

P10 l19: the annual mean

P10 l23: small bias of total column O3? Clarify.

P10 l31: 20 N/S -> 20N – 20S

P11 It should be made clear that there is a large difference between a "Radiative Effect" and a "Radiative Forcing".

P11 l12: "Nevertheless" seems misleading. Just because the present-day RE is small it doesn't follow that the RF is small. Indeed the tropospheric O3 RF (pre-industrial to present-day) peaks in the sub-tropics.

P11 l16: As already mentioned – we've already had some results.

P11 l19 . . .shows modelled annual. . .

P11 l20-21: We present results from adding. . .

P11 l23: "expected" sounds a bit presumptive. A similar pattern to that found previously?

P11 l27: chemistry -> chemical

P12 l19: (non-lightning) climate-induced

P13 l16: weighted by latitudinal area (?)

P13 l24: Clarify the origin of the quoted plus/minus value. Is it +/- 1SD from interannual variability? (as stated p10 l25) If so, please say explicitly how you calculated this – 10 years of data from run 1 and 10 years of data from run 2 – is it just based on the difference between years 1, 2, etc.? Or is it something more sophisticated? I'm not quite sure what this value really represents, and you use it to justify the significance of particular results later, so it should be clear.

P14 l7: partly associated

P14 l17: larger -> higher

P14 l31: . . .distribution of changes. . .

P15 l7: ...global forcing between 2000 and 2100... This seems to be the combined RF of CLIMATE+LIGHTNING++O3-RECOVERY+++METHANE. But does it include changes in (other) anthropogenic emissions (i.e. NOx, CO, etc.)? Clarify.

P15 l21: NB changes in the methane lifetime occur on OH timescales (seconds). Impacts on CH4 concentrations are felt on decadal timescales. Rephrase.

P15 l25 It would be better to quote these imposed CH4 concentrations in Section 2.2.

P15 l28: ...global mean methane...

P16 l19-20: Clarify – is this small negative forcing just the long-term CH4-related O3 component (I think so?) or the (CH4+O3) net RF?

P16 l23: I'm struggling to find the corresponding ACCMIP value in Stevenson et al – specify the table or figure in that paper? Did ACCMIP report directly equivalent results?

P18 l16: Clarify – are you changing anthropogenic emissions? Also, it must be really clear that whenever you say 2100 you mean 2100 under the RCP8.5 scenario.

P19 l13: I think the number in Stevenson et al. (2013) is 600 +/- 30% (i.e. 180) W/m2?

P19 22: Is a whole column NRF sensible? The large difference relative to Gauss a few lines later reinforces that it probably is not sensible.

P20 l5: leaves -> alone produces a small positive...?

P20 l10: which may be -> which are?

P20 l31: RCP8.5 scenario

P21 l10-14: So the global average column O3 is OK, just its spatial distribution isn't?

Table 1: What happens to anthropogenic trace gas emissions? Do they all follow RCP8.5 or are they kept at 2000? Clarify: by 2100 you mean 2100 RCP8.5 (for climate and CH4) – but what scenario for ODS?

Table 2: Possibly clarify CLIMATE does not include any climate-change related changes in LNOx.

Table 3: Clarify that these are RFs for 2100 RCP8.5 relative to year 2000.

Figure 2: It took me a little while to work out that (a) shows CLIMATE-CNTRL; (b) shows +LIGHTNING-CLIMATE; (c) shows ++O3-RECOVERY - +LIGHTNING; and (d) shows +++METHANE - ++O3-RECOVERY. Is that correct? This should somehow be made clearer.

Figure 3: The ZM right panels would probably be better if they all shared the same x-axis scale. Also the dashed and dotted grey lines on these are hard to see.

Figure 4: Units are W m-2 / DU.

Figure 5: I was confused by the extra 1750-2000 overall forcing on the total bar. Also clarify that the total bars are simple sums of the four bars above – is that right? What about changes in anthropogenic emissions 2000-2100 RCP8.5? Are they somehow included here, or definitely not? I'm confused.

---

## Short Comment (SC1) · 25 Jan 2018

On behalf of myself and Dr A. C. Maycock - We thank the authors of this study for providing additional process-based estimates of future ozone stratosphere-adjusted radiative forcing (RF) using simulations from the WACCM model. This well complements our own study with a different model (Banerjee et al., 2017). While comparing our two studies, we noted some pertinent issues that should be addressed in this manuscript:

-We ask the authors to clarify the tropopause height used throughout the manuscript. Has the chemical tropopause (150 ppbv O3) been used to separate stratospheric and tropospheric ozone (as suggested by the caption of Figure 2)? Has the radiative kernel been computed using the 200 hPa level as a tropopause (as in Rap et al. (2015))? If

so, the authors should consider maintaining a consistent definition of the tropopause across all their calculations, or at the very least testing the sensitivity of the results to this choice.

-A main assumption in utilizing the radiative kernel is that the RF scales linearly with the perturbation. This assumption of linearity has been shown to hold for tropospheric ozone perturbations (Rap et al., 2015). However, the same might not necessarily be true for stratospheric ozone perturbations, for which the stratospheric temperature adjustment is an important component of the RF and one that might introduce non-linearities. A simple test of linearity would be to compare the results obtained using the radiative kernel to an RF calculation using the full O3 perturbation (e.g. for CLIMATE-CNTRL).

-During the review process, we performed further calculations that show only a small sensitivity of the total RF, and separate stratospheric and tropospheric ozone RFs, under climate change at RCP4.5 and 8.5 to climate-driven changes in tropopause height; i.e. using the control versus scenario-consistent tropopause height, with the latter being higher under a warmer climate. If possible, we ask the authors to also test and report this sensitivity.

-Highlighting and understanding inter-model differences/similarities is important in constraining the future ozone RF. A key difference between our two studies is mentioned on P20L16. However, we would also like the authors to highlight a key similarity, and hence the robust result, that the stratospheric ozone changes under future ODS reductions ultimately drive almost 100% of the tropospheric ozone RF.

References

Banerjee, A., Maycock, A. C. and Pyle, J. A.: Chemical and climatic drivers of radiative forcing due to changes in stratospheric and tropospheric ozone over the 21st century, Atmos. Chem. Phys. Discuss., 15(x), 1–19, doi:10.5194/acp-2017-741, 2017.

Rap, A., Richards, N. A. D., Forster, P. M., Monks, S. A., Arnold, S. R. and Chipperfield, M. P.: Geophysical Research Letters, Geophys. Res. Lett., 42, 5074–5081, doi:10.1002/2015GL063354.
* * *

---

## Referee Comment (RC2) · Anonymous Referee #2 · 26 Jan 2018

Review of 'Key drivers of ozone change and its radiative forcing over the 21st century' by Iglesias-Suarez et al.

This paper attempts to evaluate the contributions of different drivers in the radiative forcing of future ozone. The specific factors are expected future changes in climate, ODS levels, and methane levels for the worse-case scenario pathway (RCP8.5). The ozone changes are calculated with a chemistry-climate model (CESM1-WACCM) forced with different elements of the RCP8.5 scenario. The radiative forcings of the model-calculated ozone fields are estimated with an off-line version of a well-established radiative transfer model. The stratospherically adjusted radiative forcing is calculated using the fixed dynamic heating approximation. The authors estimated using a radiative transfer model authors find that the large methane increase planned

in RCP8.5 is the most important factor among these 3 factors. The paper is reasonably well written. The results are interesting and relevant to JGR community. They will be very useful to scientists studying the climate impact of ozone changes and their drivers. Nonetheless, the authors can certainly improve the presentation of their work. The two parts that need to be improved are the introduction and the description of the simulations. The sections on results and conclusions are satisfactory. Overall, I recommend publication after a number of minor points are addressed. This would substantially improve the clarity of the paper.

In the introduction, the authors should state explicitly the important drivers of ozone that are covered here. They could discuss more extensively these key drivers and, more importantly, how important they might be for radiative forcing. For instance, the changes in anthropogenic emissions, notably emissions of ozone precursors, have been and will be fundamental for changes in tropospheric ozone. There are also quite a few useful papers that provide estimates of the radiative forcing from tropospheric or stratospheric ozone changes (including works from some of the co-authors) that could be cited. This will give some ideas about the significance of the radiative forcings calculated here. p3, l3: There is something missing sentence to link and introduce the second sentence. Perhaps, However, tropospheric ozone is also significantly affected by the change in UV reaching the troposphere brought about by the ticker stratospheric ozone layer . . . l29, p3: "in the lower stratosphere (through enhanced heterogeneous ozone destruction)". It is certainly the case in the polar regions, but not the tropics. Add 'polar'.

p1, p4: "associated to an increase of relatively ozone-poor air entering from the troposphere". It is a misunderstanding. The loss in tropical lower stratospheric ozone has nothing to do with ozone-poor air entering the tropical stratosphere. It is the fact that air is moving faster and so less ozone is produced. The he concentration of ozone in the tropical pipe is determined by the ascent rate and mixing and not by the initial concentration at the tropical tropopause which is in effect extremely small compared to

stratospheric values. I suggest that the authors read Avallon and Prather, JGR, 1996

l12, p4: A reference for this value should be provided.

l31, p4: add 'tropospheric'

l1, p5: Rephrase. Perhaps diagnose the contribution of change in ozone. . .

l32, p5: 'processed-based' sounds good. But I don't know what it means because there is no explanation.

l4, p6: I don't think that they have just identified the forcing.

l3-l30, p6: somewhere, it should be stated explicitly which ozone drivers are not considered and whether they are important for radiative forcing.

l14, p6: "provide a gauge". Do it mean estimate? if yes, why not use estimate.

l4-8, p7: Add that it is a chemistry-climate model.

l26-32, p7: A bit confusing. Do you first run the stratospheric ozone tracer O3S without deposition and then you modify the O3S output fields by removing some of it based on an additional run where the deposition mass fluxes are calculated and stored?

l6, p8: numerical experimental set up or modelling set up.

l14-16, p8: The emissions are fixed so the importance of this driver for tropospheric ozone and radiative forcing is not explored. I was not sure up to that point.

Section 2.2, p8-9: There is a table provide about the list of runs but there is no explanation and rational provided about the runs CLIMATE, LIGHTNING, O3-RECOVERY, and METHANE. The reader has to guess but it can be confusing. Can the authors explain the different runs and the reasoning behind the choice of these runs?

l7-9, p10: The Tilmes et al paper states: Tropospheric column ozone is reproduced within +/-10 DU of the observations, with a close agreement to the satellite climatology within less than +/- 5 DU in low and mid-latitudes in spring and summer. Add in spring

and summer.

l22, p10: add tropospheric

l9, p11: "Constrained"? do you mean confined. l8, p16: it is at the upper end, not mid-upper.

l16, p16: It should be pointed that this estimation assumes that the relationships between changes in methane, ozone and radiative forcing are linear.

---

## Author Comment (AC1)

**Response to reviewers and short comments – Key drivers of ozone change and its radiative forcing over the 21st century**

We are grateful for the feedback of the two reviewers and A. Banerjee. We hope their comments and concerns are addressed below. Our responses (i.e. changes and information) follow each comment in **blue**.

Clarification of the drivers explored in the study, as requested by the two reviewers, has been an important revision of the manuscript. Furthermore, in the process of revising the paper following A. Banerjee's short comment, an issue with the radiative kernel (RK) calculation was identified related to the convergence of the stratospheric temperature adjustment. This has since been corrected and found to have only a very small effect on the calculated ozone radiative forcings, but it has a more substantial effect on the diagnosed ozone radiative effect for near present day shown in Figure 2 (all figures are attached). The calculations have been updated in the revised manuscript to reflect the corrected RK, but the conclusions of the study regarding future ozone radiative forcing are unchanged.

**Responses to Reviewer #1**

**(a) General comment**:

The paper is about modelled ozone changes between 2000 and 2100, but after reading it I am unclear how the major influence of changing anthropogenic emissions is included (the focus is on changes in climate, lightning, ozone-depleting substances, and methane).

**Response**: This comment has also been picked up by Reviewer #2 (see below). We think that it should be clear which ozone drivers are/and are not consider in this study. The last paragraph of the introduction (Sect. 1) has now been expanded:

"… **Note this study does not address anthropogenic reductions in $NO_x$ and non-methane volatile organic compounds emissions, since they play a marginal role in future ozone RF under the RCP8.5 scenario (based on an additional simulation not presented here)**."

This has also been made clear in other sections (e.g. Sect. 2.2). Please, see also the marked-up manuscript.

**(b) Specific comments**:

**Page 2, Lines 4–6**. Do changes in the stratosphere account for 47% of the overall RF for both 1750-2000 and 2000-2100?

Response: Changes in stratospheric-produced ozone refers to the overall radiative forcing (RF) "in this set of simulations". Therefore, it includes only the 2000–2100 period. The sentence has been rewritten:

"Changes in stratospheric-produced ozone account for **~50 %** of the overall radiative forcing **for the 2000–2100 period** in this set of simulations…"

**Page 2, Line 15**. 90% by mass.

Response: Fixed. Thanks.

**Page 2, Line 19**. It is difficult to agree that ozone is 'the main' concern for climate change and air quality (CO2 and PM are probably more important!) – suggest change to 'an important'.

Response: We agree that 'an important' may be more appropriate than 'a main'. Fixed. Thanks.

**Page 2, Line 26**. Delete comma after although.

Response: Fixed. Thanks.

**Page 2, Lines 30**. "BDC is the wave-driven factor" – rephrase.

Response: The sentence has now been rephrased:

"The BDC **governs the meridional**…"

**Page 3, Line 13**. ODS (not ODSs).

**Response**: Fixed. Thanks.

**Page 3, Line 18**. Troposphere (not stratosphere).

**Response**: Higher levels of ozone in the stratosphere can affect tropospheric ozone (e.g. abundance and distribution) due to both an enhanced stratosphere-troposphere exchange and changes in photolysis rates. We describe how stratospheric ozone may affect ozone in the troposphere and not discuss the actual change. We did refer in P3L18 to the 'stratosphere' and not the troposphere.

**Page 3, Lines 21–24**. Clarify – do reductions in O3 photolysis mean more tropospheric O3?

**Response**: We point out that the impacts of reductions in tropospheric photolysis rates are complex and have an indirect feedback on ozone via changes in methane lifetime. This is relevant since long-term tropospheric ozone radiative forcing associated with changes in OH and methane lifetime is explored. The sentence has now been rewritten to clarify the above comment:

"**In turn**, reductions in ozone photolysis result in lower OH concentrations – i.e. ... – and therefore longer methane lifetime**, with consequences for long-term tropospheric ozone abundances** (e.g. Morgenstern et al., 2013; Zhang et al., 2014)."

**Page 4, Line 1**. to -> with.

**Response**: Fixed. Thanks.

**Page 4, Line 4**. lower -> reduced.

**Response**: Fixed. Thanks.

**Page 4, Line 9**. What about aircraft NOx?

**Response**: Indeed, emissions from aircrafts are an important source of $NO_x$ away from the surface. However, the authors refer only to natural emissions of $NO_x$ and not those from anthropogenic sources. Since changes in anthropogenic emissions of ozone precursor species – other than methane – are not explored in the manuscript (addressed above in (a) and in Responses to Reviewer #2), we believe is not relevant mentioning aircraft $NO_x$ emissions (i.e. this is fixed at year 2000 in all simulations).

**Page 5, Lines 15–17**. Aren't higher latitudes than the tropics more strongly influenced by stratospheric influx of O3?

**Response**: The authors agree that changes in stratospheric influx of ozone are dependant on the driver. For example, greater changes in stratospheric ozone influx at high latitudes compared to the tropics have been shown for ozone recovery (i.e. reduced ODS concentrations), whereas the opposite is true in a warmer climate (e.g. Fig.1a,c,e in Banerjee et al., 2016 – doi.org/10.5194/acp-16-2727-2016). However, this is a good point and we have now rewritten the sentence:

"…, a region greatly influenced by **changes in** stratospheric influx (e.g. Hegglin and Shepherd, 2009; Zeng et al., 2010**; Banerjee et al., 2016**) and lightning-produced ozone (e.g. **Banerjee et al., 2014;** Liaskos et al., 2015) **in a warmer climate**."

**Page 5, Lines 27–29**. Sentence unclear – reword.

**Response**: The sentence has now been rewritten:

"**Note that the methane concentration in 2100 is more than double that in the year 2000 following the RCP8.5 emissions scenario**."

**Page 6, Line 8**. i.e. -> e.g. (also following lines 9 and 11).

**Response**: Fixed. Thanks.

**Page 6, Lines 19–23**. Unclear – reword.

**Response**: The sentence has now been rewritten:

"**We explore the robustness of the ozone radiative forcings associated with the above drivers under different background conditions due to non-linearities in ozone responses**."

**Page 7, Lines 19**. Is CH4 an ODS? (I don't think so).

**Response**: Methane is not an ODS. The word 'other' has now been removed from the sentence to avoid misunderstanding.

**Page 7, Lines 27–28**. It may have been recommended by CCMI, but I don't really understand why the O3S tracer can't also be lost by O3 dry deposition (like any ozone molecule). Can you clarify why? This seems like an odd approach. I see you apply a correction factor – I guess to account for this.

**Response**: We agree the O3S tracer should be lost by dry deposition like any other ozone molecule. Since the model configuration used for the experiments presented in the study followed the CCMI recommendation, we included dry deposition of O3S in an additional run – as guessed by the two reviewers (see below) – to derive a correction factor that account for this. The sentence has now been rewritten for better clarification:

"**To account for dry deposition of O3S, we apply an annual global correction factor based on an additional model simulation (not used in the main results)**."

**Page 8**. (This should be somewhere in the model description or experimental set-up): Is ozone coupled radiatively to the climate model? I assume it is, and this means that any changes in ozone generate changes in meteorology. This should be made clear, as it has important implications for how the results are interpreted.

**Response**: Radiatively-active chemical species, such as ozone, are coupled to the radiation scheme and may therefore affect meteorology, as explicitly indicated in the manuscript (see Page 7, Lines 18–20 in the discussion manuscript).

**Page 8, Line 10**. There is a minor inconsistency in your naming/approach – 1990-2009 is used to represent 2000; 2080-2099 is used to represent 2100. Why not nominally 2090? Actually on the next line you say each experiment is 20 years with only the last 10 years used – so 2000-2009 or the 2000s and 2090-2099 i.e. the 2090s?

**Response**: We agree there is a minor inconsistency between the climatology of the sea surface temperatures and sea ice concentrations used to drive the year 2000 and 2100 experiments. Using 20 year averages centred on the target year would be ideal (i.e. 1990–2009 period for year 2000). However, since SST information for the 2090–2109 period was not available for the model, the last 20 years of the 21$^{st}$ century were used to construct the climatology for the year '2100'. However, the concentrations of atmospheric gases applied in the perturbation experiments are taken from year 2100 in the RCP8.5 scenario, and hence we use the naming convention '2100'. Please note that time slice simulations use the same boundary conditions run repeatedly (i.e. cyclical). Here, each simulation is allowed to spin-up for 10 years and are run for another 10 years, which were used in the main results. For example to explore ozone recovery, ODS concentrations would be fixed at year 2100 and everything else would remain at year 2000. This idealise condition is run repeatedly for 20 years, but the analysis only uses the last 10 years (i.e. after the model reached 'equilibrium').

We think that keeping nominal changes to year 2100 really helps the broader audience follow the story and take up the main messages. Nevertheless, the paragraph has now been rewritten to address the above concern:

"**An average over** 1990-2009 **is used** to represent the year 2000**; since the existing model simulation did not cover the period 2090-2109, an average over** 2080-2099 **is used** to represent **conditions at** the end of the 21st century (nominally 2100). **Note, however, that the perturbed concentrations of atmospheric gases are taken from year 2100 in the RCP8.5 scenario, and hence these experiments are labelled as 2100 in the manuscript**."

**Page 8, Line 14**. You use MACCity anthropogenic emissions, but don't make it clear if they are held fixed at year 2000 levels or if they change between 2000 and 2100 (RCP8.5). This needs to be clear as it is also very important. **Page 8, Line 23**. Clarify

if land-use is changing in these runs – if so this would have implications for, e.g., BVOC emissions, etc.

**Response**: A sentence has now been included to address the above concerns:

"**Changes in ozone precursors – other than CH$_4$ – and land-use changes are not explored here (i.e. these are fixed at year 2000 levels in all experiments).**"

**Page 8, Line 19**. Phases.

**Response**: Fixed. Thanks.

**Page 10, Line 4**. I think Section 3 belongs within the "Results", rather than prior to it. It contains some results! **Page 11, Line 16**. As already mentioned – we've already had some results.

**Response**: We agree that Section 3 does have some results – i.e. present-day ozone radiative effect (RE) both observed and modelled using the RK method. Fixed. Thanks.

**Page 10, Line 7**. "a 25 %".

**Response**: Fixed. Thanks.

**Page 10, Line 19**. "the annual mean".

**Response**: Fixed. Thanks.

**Page 10, Line 31**. 20 N/S -> 20N – 20S.

**Response**: Fixed P10L31 and elsewhere. Thanks.

**Page 11**. It should be made clear that there is a large difference between a "Radiative Effect" and a "Radiative Forcing".

**Response**: This is already addressed in the introduction (P5L6–7). However, this is a good comment. A sentence has now been included at the end of Sect. 3.1 to reinforce

a clear difference between RE and RF and link it with subsequent sections (ozone changes and RF):

"**Note the RE is the radiative flux imbalance at a given time due to a radiatively active species (e.g. with and without ozone), whereas the RF refers to the change in RE over time**."

**Page 11, Line 12**. "Nevertheless" seems misleading. Just because the present-day RE is small it doesn't follow that the RF is small. Indeed the tropospheric O3 RF (pre-industrial to present-day) peaks in the sub-tropics.

**Response**: We agree this is a fair comment. For example, changes in ozone are very efficient in affecting the radiative flux at low latitudes. The sentence has now been rewritten to address the above concern:

"**Although** tropical and subtropical regions are of particular interest for future changes in ozone and its resulting radiative forcing **(i.e. highest radiative efficiency), there is a large NH/SH compensation as shown by the annual and global mean forcings**."

**Page 11, Line 19**. "… shows modelled annual…".

**Response**: Fixed. Thanks.

**Page 11, Lines 20–21**. "We present results from adding…".

**Response**: Fixed. Thanks.

**Page 11, Line 23**. "expected" sounds a bit presumptive. A similar pattern to that found previously?

**Response**: Fixed. Thanks.

**Page 11, Line 27**. chemistry -> chemical.

**Response**: Fixed. Thanks.

**Page 12, Line 19**. "(non-lightning) climate-induced".

**Response**: Fixed. Thanks.

**Page 13, Line 16**. Weighted by latitudinal area (?).

**Response**: The sentence has now been rewritten to address the above comment:

"Note that zonal mean forcings are **weighted by latitudinal area (i.e. cosine-latitude)**…"

**Page 13, Line 24**. Clarify the origin of the quoted plus/minus value. Is it +/- 1SD from inter- annual variability? (as stated p10 l25) If so, please say explicitly how you calculated this − 10 years of data from run 1 and 10 years of data from run 2 − is it just based on the difference between years 1, 2, etc.? Or is it something more sophisticated? I'm not quite sure what this value really represents, and you use it to justify the significance of particular results later, so it should be clear.

**Response**: The sentence has now been rewritten to clarify the significance used:

"The global forcing associated with **climate (Clm−Cnt; Fig. 4a) of −70 ± 102 mWm$^{-2}$ is relatively** small and not highly statistically significant **(errors denote 1 standard error associated with the 10 year interannual variability of ozone change unless otherwise specified)**."

**Page 14, Line 17**. "partly associated"; larger -> higher.

**Response**: Fixed. Thanks.

**Page 14, Line 31**. "… distribution of changes…"

**Response**: Fixed. Thanks.

**Page 15, Line 7**. "… global forcing between 2000 and 2100…" This seems to be the combined RF of CLIMATE+LIGHTNING++O3-RECOVERY+++METHANE. But

does it include changes in (other) anthropogenic emissions (i.e. NOx, CO, etc.)? Clarify.

**Response**: As commented elsewhere (General Comments for Reviewers #1-2) and included in the revised manuscript, this study does not explore ozone RF associated with changes in anthropogenic emissions of ozone precursors other than $CH_4$.

**Page 15, Line 21**. NB changes in the methane lifetime occur on OH timescales (seconds). Impacts on CH4 concentrations are felt on decadal timescales. Rephrase.

**Response**: The sentence has now been rephrase to clarify the above comment:

"Future climate change and emissions of ODSs and methane will affect the oxidising capacity of the atmosphere (e.g., via hydroxyl radicals, OH), which influences the methane lifetime ($\tau CH4$) **and its concentration. In turn, changes in methane concentrations result in a "long-term" response of tropospheric ozone at decadal time scales**…"

**Page 15, Line 25**. It would be better to quote these imposed CH4 concentrations in Section 2.2.

**Response**: Indeed, imposed $CH_4$ concentrations were already specified in Table 1. Therefore, these concentrations have now been removed from the sentence. Thanks.

**Page 15, Line 28**. "… global mean methane…"

**Response**: Fixed. Thanks.

**Page 16, Lines 19–20**. Clarify – is this small negative forcing just the long-term CH4-related O3 component (I think so?) or the (CH4+O3) net RF?

**Response**: Yes, the paragraph refers only to long-term tropospheric ozone RFs associated with methane feedbacks. The sentence has now been rewritten to clarify the above concern:

"…which results in a small negative forcing of −19 mWm$^{-2}$ **due to the methane feedback**…"

**Page 16, Line 23**. I'm struggling to find the corresponding ACCMIP value in Stevenson et al – specify the table or figure in that paper? Did ACCMIP report directly equivalent results?"

**Response**: The authors provide a corresponding ACCMIP value from Table 8 (first row, middle column for each model ozone radiative forcing) in Stevenson et al. (2013), when considering the same change in methane concentrations. Note in Stevenson et al. (2013) ozone RFs due to methane feedback are for the pre-industrial period (1850–2000), when methane concentrations increased by ~960 ppbv. Here, we consider 21$^{st}$ century (2000–2100) methane concentration change of ~ 2000 ppbv following the RCP8.5. Because there are no directly equivalent results from ACCMIP, we simply extrapolated ACCMIP values. The sentence has now been rewritten to help the reader understand the comparison above commented:

"This forcing is within the range of ~ **40–120 (mean value of 60) mWm$^{-2}$ from the ACCMIP ensemble (Table 8 in Stevenson et al., 2013), when considering the same change in methane concentrations (note their values have been linearly extrapolated)**."

**Page 18, Line 16**. Clarify – are you changing anthropogenic emissions? Also, it must be really clear that whenever you say 2100 you mean 2100 under the RCP8.5 scenario.

**Response**: The sentence has now been rewritten to clarify the emission scenario followed:

"We calculate a net ozone radiative forcing of 435 ± 108 mWm$^{-2}$ corresponding to the year 2100 **under the RCP8.5 emissions scenario** compared to present-day…"

**Page 19, Line 13**. I think the number in Stevenson et al. (2013) is 600 +/- 30% (i.e. 180) W/m2?

**Response**: Stevenson et al. (2013) and Skeie et al. (2011) adopt an overall uncertainty of ±30 % representing the 95 % confidence interval, as we do later in the manuscript. P19L13 refers to the model range uncertainty in Stevenson et al. (2013), which is ±20 %. Nevertheless, this is a good observation and we have now included the latter uncertainty in the main text. Thanks.

**Page 19, Line 22**. Is a whole column NRF sensible? The large difference relative to Gauss a few lines later reinforces that it probably is not sensible.

**Response**: In the present study is difficult to assess whether a whole column NRF would be sensible. Indeed, there is a relatively large difference compared to Gauss et al. (2003) due to ozone changes in the upper stratosphere (not included in the latter) have a relatively small impact on RF. Therefore, we have now removed the net ozone NRF from Sect. 5.

**Page 20, Line 5**. "leaves -> alone produces a small positive…?"

**Response**: Fixed. Thanks.

**Page 20, Line 10**. "which may be -> which are?"

**Response**: Fixed. Thanks.

**Page 20, Line 31**. RCP8.5 scenario.

**Response**: Fixed. Thanks.

**Page 21, Lines 10–14**. So the global average column O3 is OK, just its spatial distribution isn't?

**Response**: The sentence has now been rewritten to clarify the above comment:

"The fact that **similar spatial distribution** biases are apparent in many climate models…"

**Table 1**. What happens to anthropogenic trace gas emissions? Do they all follow RCP8.5 or are they kept at 2000? Clarify: by 2100 you mean 2100 RCP8.5 (for climate and CH4) – but what scenario for ODS?

**Response**: Table 1 has now been updated to clarify that 'climate' and 'CH$_4$' for year 2100 follow the RCP8.5 emissions scenario, and 'ODSs' follow the halogen scenario A1, which was already included in the main text (Sect. 2.2).

**Table 2**. Possibly clarify CLIMATE does not include any climate-change related changes in LNOx.

**Response**: Table 2 has now been updated to clarify that the Clm and Cnt+fLNOx simulations used fixed lightning-induced NO$_x$ emissions from the Cnt run. Thanks.

**Table 3**. Clarify that these are RFs for 2100 RCP8.5 relative to year 2000.

**Response**: Fixed. Thanks.

**Figure 2**. It took me a little while to work out that (a) shows CLIMATE-CNTRL; (b) shows +LIGHTNING-CLIMATE; (c) shows ++O3-RECOVERY - +LIGHTNING; and (d) shows +++METHANE - ++O3-RECOVERY. Is that correct? This should somehow be made clearer.

**Response**: Yes, it is correct. We agree this should be made clearer (also addressed by Reviewer #2). We have now rewritten some parts of the main text (see marked-up manuscript) as well as captions of Table 3 and Figures 2–5.

**Figure 3**. The ZM right panels would probably be better if they all shared the same x-axis scale. Also the dashed and dotted grey lines on these are hard to see.

**Response**: Fixed. Thanks.

**Figure 4**. Units are W m-2 / DU.

**Response**: Fixed. Thanks.

**Figure 5**. I was confused by the extra 1750-2000 overall forcing on the total bar. Also clarify that the total bars are simple sums of the four bars above – is that right? What about changes in anthropogenic emissions 2000-2100 RCP8.5? Are they somehow included here, or definitely not? I'm confused.

**Response**: Caption of Figure 6 (former Figure 5) has now been rewritten to address the above concerns:

"… The overall ozone forcing (Total) is **the sum of the individual forcings (Climate, Lightning, O$_3$-recovery and Methane from Table 3)** scaled to 1750 (star-hatched). Dots and error bars indicate the mean and the 95 % confidence intervals of the forcings respectively. **The information on pre-industrial ozone forcing (1750–2000) and sources of uncertainty are detailed in Sect. 4**."

**Responses to Reviewer #2**

**(a) General comment**:

In the introduction, the authors should state explicitly the important drivers of ozone that are covered here. They could discuss more extensively these key drivers and, more importantly, how important they might be for radiative forcing. For instance, the changes in anthropogenic emissions, notably emissions of ozone precursors, have been and will be fundamental for changes in tropospheric ozone. There are also quite a few useful papers that provide estimates of the radiative forcing from tropospheric or stratospheric ozone changes (including works from some of the co-authors) that could be cited. This will give some ideas about the significance of the radiative forcings calculated here.

**Response**: We have now stated explicitly in the introduction (Sect. 1) and the experimental design (Sect. 2.2) the different drivers that are covered in this study. We have also discussed the drivers that are not addressed here (i.e. non-methane anthropogenic ozone precursors) based on their importance for ozone forcing under the RCP8.5 emissions scenario. In addition, we have now included in the introduction previous modelling estimates of future tropospheric ozone forcing to put into context the potential role of ozone as radiative active species and the RFs calculated here compared to the total RF of the RCP8.5 emissions scenario.

**(b) Specific comments**:

**Page 3, Line 3**. There is something missing sentence to link and introduce the second sentence. Perhaps, However, tropospheric ozone is also significantly affected by the change in UV reaching the troposphere brought about by the ticker stratospheric ozone layer…

**Response**: We think this comment refers to Page 3 Lines 22–24. The sentence has now been rewritten to address the above comment:

"**However, tropospheric ozone is also significantly affected by the change in ultraviolet radiation reaching the troposphere brought about by the ticker stratospheric ozone layer. In turn**, reductions in ozone photolysis result in lower OH concentrations…"

**Page 3, Line 29**. "in the lower stratosphere (through enhanced heterogeneous ozone destruction)". It is certainly the case in the polar regions, but not the tropics. Add 'polar'.

**Response**: Fixed. Thanks.

**Page 4, Line 4**. "associated to an increase of relatively ozone-poor air entering from the troposphere". It is a misunderstanding. The loss in tropical lower stratospheric ozone has nothing to do with ozone-poor air entering the tropical stratosphere. It is the fact that air is moving faster and so less ozone is produced. The he concentration of ozone in the tropical pipe is determined by the ascent rate and mixing and not by the initial concentration at the tropical tropopause which is in effect extremely small compared to stratospheric values. I suggest that the authors read Avallon and Prather, JGR, 1996.

**Response**: We agree that the sentence as it was written is misleading. The sentence has now been rewritten to address the above comment:

"… which results in (i) decreases in tropical lower stratospheric ozone, associated with **a relatively faster ventilation and reduced ozone production (Avallone and Prather, 1996)**; and (ii) ozone increases in the upper troposphere, particularly in the region of the subtropical jets, linked to the descending branch of the BDC **(e.g. Kawase et al., 2011; Banerjee et al., 2016)**…"

**Page 4, Line 12**. A reference for this value should be provided.

**Response**: Please note that Schumann and Huntrieser (2007) estimated annual and global lightning-induced $NO_x$ emissions, as well as provided a range of sensitivities to climate change previously reported from different chemistry-climate models (see their Table 14 and main text). We have now updated the above reference to address the above concern:

"… (Schumann and Huntrieser, 2007**, and references therein**)."

**Page 4, Line 31**. add 'tropospheric'.

Response: Fixed. Thanks.

**Page 5, Line 1**. Rephrase. Perhaps diagnose the contribution of change in ozone.

Response: We have now rewritten the sentence:

"… diagnose the **contribution of ozone changes** on the atmospheric radiative budget."

**Page 5, Line 32**. 'processed-based' sounds good. But I don't know what it means because there is no explanation.

Response: We have now rewritten the sentence to explain what 'processed-based' means in this study:

"…in a processed-based approach **– i.e. imposing one single forcing at a time** (Collins et al., …"

**Page 6, Line 4**. I don't think that they have just identified the forcing.

Response: The sentence has now been rewritten to address the above comment:

"Other modelling studies focused on the radiative effects of **tropospheric (e.g. Gauss et al., 2003; Stevenson et al., 2013) and stratospheric (Bekki et al., 2013)** ozone changes under future emission scenarios in a non processed-based fashion. One study has recently identified the indirect tropospheric and stratospheric ozone RF between 2000 and 2100 due to individual perturbations (**Banerjee et al., 2018**)."

**Page 6, Lines 3–30**. Somewhere, it should be stated explicitly which ozone drivers are not considered and whether they are important for radiative forcing.

Response: This comment has also been picked up by Reviewer #1 (see above). We agree that it should be stated explicitly which ozone drivers are and are not consider. The last paragraph of the introduction (Sect. 1) has now been expanded:

"… We use the Community Earth System Model (CESM1) in its "high-top" (up to 140 km) atmosphere version – the Whole Atmosphere Community Climate Model (WACCM) – and a series of sensitivity simulations to quantify the radiative effects of ozone due to **(1)** climate change, **(2) lightning-induced NO$_x$ emissions, (3)** stratospheric ozone recovery, and **(4)** methane emissions between 2000 and 2100 **following the RCP8.5 emissions scenario. We explore the robustness of the ozone radiative forcings associated with the above drivers under different background conditions due to non-linearities in ozone responses.** Moreover, here we use a synthetic ozone tracer to unambiguously identify stratospheric- and tropospheric-produced ozone forcing. **Note this study does not address anthropogenic reductions in NO$_x$ and non-methane volatile organic compounds emissions, since they play a marginal role in future ozone RF under the RCP8.5 scenario (based on an additional simulation not presented here).**"

**Page 6, Line 14**. "provide a gauge". Do it mean estimate? if yes, why not use estimate.

**Response**: Fixed. Thanks.

**Page 7, Lines 4–8**. Add that it is a chemistry-climate model.

**Response**: The sentence has now been rewritten as follows:

"We use the CESM **(version 1.1.1) chemistry-climate model** with a configuration…"

**Page 7, Lines 26–32**. A bit confusing. Do you first run the stratospheric ozone tracer O3S without deposition and then you modify the O3S output fields by removing some of it based on an additional run where the deposition mass fluxes are calculated and stored?

**Response**: We agree the O3S tracer should be lost by dry deposition like any other ozone molecule. Since the model configuration used for the experiments presented in the study followed the CCMI recommendation, we included dry deposition of O3S in an additional run – as guessed by the two reviewers (see below) – to derive a

correction factor that account for this (see also Reviewer #1, Page 7, Lines 27–28). The sentence has now been rewritten for better clarification:

"**To account for dry deposition of O3S, we apply an annual global correction factor based on an additional model simulation (not used in the main results)**."

**Page 8, Line 6**. "numerical experimental set up or modelling set up."

**Response**: Fixed. Thanks.

**Page 8, Lines 14–16**. The emissions are fixed so the importance of this driver for tropospheric ozone and radiative forcing is not explored. I was not sure up to that point.

**Response**: This is a good point. Since anthropogenic emissions of ozone precursors (other than methane) are not explored, we have now removed their description in the sentence. Thanks.

**Section 2.2, Pages 8–9**. There is a table provide about the list of runs but there is no explanation and rational provided about the runs CLIMATE, LIGHTNING, O3-RECOVERY, and METHANE. The reader has to guess but it can be confusing. Can the authors explain the different runs and the reasoning behind the choice of these runs?

**Response**: We have now explained in more detail the list of runs and the reasoning behind the experimental design:

"… The control simulation (Cnt) had all boundary conditions set to the year 2000**. Then each sensitivity simulation added one single driver (i.e. boundary condition changed to the year 2100) at a time. For example, while the climate-related ozone RF (with fixed LNO$_x$ emission) is explored comparing the Clm−Cnt simulations, the forcing associated with changes in lightning-induced NO$_x$ emissions is quantified comparing the Lnt−Clm simulations, and so forth. This method provides a different estimate of the overall net ozone RF compared to exploring the impact of the individual drivers alone (e.g. it accounts for non-linear effects**

**that may be neglected by exploring each perturbation compared to the reference simulation). However, since the attribution of forcings to individual drivers may be sensitive to different background conditions, we also evaluate the robustness of the experimental design (see Sect. 3.5).**"

**Page 10, Lines 7–9**. The Tilmes et al paper states: Tropospheric column ozone is reproduced within +/-10 DU of the observations, with a close agreement to the satellite climatology within less than +/- 5 DU in low and mid-latitudes in spring and summer. Add in spring and summer.

**Response**: Fixed. Thanks.

**Page 10, Line 22**. Add tropospheric.

**Response**: Fixed. Thanks.

**Page 11, Line 9**. "Constrained"? do you mean confined.

**Response**: Fixed. Thanks.

**Page 16, Line 8**. It is at the upper end, not mid-upper.

**Response**: Fixed. Thanks.

**Page 16, Line 16**. It should be pointed that this estimation assumes that the relationships between changes in methane, ozone and radiative forcing are linear.

**Response**: The sentence has no been expanded to address the above concern:

"**Assuming the relationships between changes in methane, ozone and radiative forcings are linear; t**he associated tropospheric ozone forcings to methane feedback…"

**Responses to A. Banerjee**

**Specific comments**:

**1**. We ask the authors to clarify the tropopause height used throughout the manuscript. Has the chemical tropopause (150 ppbv O3) been used to separate stratospheric and tropospheric ozone (as suggested by the caption of Figure 2)? Has the radiative kernel been computed using the 200 hPa level as a tropopause (as in Rap et al. (2015))? If so, the authors should consider maintaining a consistent definition of the tropopause across all their calculations, or at the very least testing the sensitivity of the results to this choice.

**Response**: We have now included a new paragraph in Sect. 2.3 to clarify the tropopause height used throughout the manuscript to separate stratospheric and tropospheric ozone RFs:

"**A chemical tropopause definition (Prather et al., 2001), using the 150 ppb ozone level of the Cnt simulation, is employed to differentiate ozone changes and associated RFs occurring in the troposphere and the stratosphere**."

In addition, we have now updated the ozone radiative kernel ($O_3$ RK; see below 2.), which was computed using a 200 hPa tropopause definition. This inconsistency introduces some level of uncertainty which has been accounted for in Sect. 4.

**2**. A main assumption in utilizing the radiative kernel is that the RF scales linearly with the perturbation. This assumption of linearity has been shown to hold for tropospheric ozone perturbations (Rap et al., 2015). However, the same might not necessarily be true for stratospheric ozone perturbations, for which the stratospheric temperature adjustment is an important component of the RF and one that might introduce non-linearities. A simple test of linearity would be to compare the results obtained using the radiative kernel to an RF calculation using the full O3 perturbation (e.g. for CLIMATE-CNTRL).

**Response**: Indeed, since the $O_3$ RK is defined as the derivative of the radiative flux relative to perturbations in ozone, the assumption of linearity is implicit in this method. Furthermore, this assumption had not been evaluated in Rap et al. (2015) for the stratosphere. Therefore, we have now updated the radiative kernel in Rap et al.

(2015) via stratospheric temperature adjustments and significantly extended Sect. 2.3 with a detailed description of the updated $O_3$ RK (including a figure with the net, long- and short-wave components). In addition, we have also evaluated the assumption of linearity for stratospheric ozone by comparing the results obtained using the O3 RK to an RF calculation using the full radiative transfer model, which has been included in the supplementary information (Figure S1). Although the use of the updated $O_3$ RK has slightly modified the resulting ozone forcings (see marked-up manuscript), the main conclusions of this work have remained.

**3**. During the review process, we performed further calculations that show only a small sensitivity of the total RF, and separate stratospheric and tropospheric ozone RFs, under climate change at RCP4.5 and 8.5 to climate-driven changes in tropopause height; i.e. using the control versus scenario-consistent tropopause height, with the latter being higher under a warmer climate. If possible, we ask the authors to also test and report this sensitivity.

**Response**: We have now tested the sensitivity of the tropospheric-stratospheric forcing partitioning associated with changes in the tropopause due to climate change under the RCP8.5 scenario and found negligible differences. This sensitivity has now been reported in the manuscript:

"… **Compared to the latter, we found a negligible difference in the partitioning of tropospheric-stratospheric forcing using a consistent chemical tropopause definition to the driver investigated (i.e. higher tropopause associated with climate change)**."

**4**. Highlighting and understanding inter-model differences/similarities is important in constraining the future ozone RF. A key difference between our two studies is mentioned on P20L16. However, we would also like the authors to highlight a key similarity, and hence the robust result, that the stratospheric ozone changes under future ODS reductions ultimately drive almost 100% of the tropospheric ozone RF.

**Response**: This is a very good point and we agree that future ODS reductions virtually drives net ozone RF. We have now included a sentence to highlight this important result:

"A striking result, however, is the contribution of the stratospheric-produced ozone to the net forcing of ~30 ± 20 % and ~99 ± 50 % due to methane and ODS concentrations respectively**, which is consistent with the findings from an independent chemistry-climate model (Banerjee et al., 2016, 2018).** This reflects the roles that methane plays in stratospheric ozone chemistry (i.e. particularly in the lower stratosphere), and that ozone recovery principally occurs in the stratosphere."

We thank again the two reviewers and A. Banerjee for their comments.

---

## Author Response (AR2)

**Response to Jens-Uwe Grooß – Key drivers of ozone change and its radiative forcing over the 21st century**

We are grateful for the feedback of Jens-Uwe Grooß. We hope his comments and concerns are addressed below. Our responses (i.e. changes and information) follow each comment in **blue**.

**Specific comments**:

**a)** In the abstract you mention the different contributions to O3 RF numbered by (1) (2), and (3) for (1) you mention "with and without lightning feedback" but you give only one number [corresponding to the sum of the first two lines of table3]. EITHER you may leave out the info "with and without lightning feedback" (the interplay between lightning-produced ozone and enhanced ozone destruction is explained later) OR you may clarify this point in the abstract.

**Response**: We agree this is indeed a good observation. We have now left out the "lightning feedback" comment and explained the uncertainty range provided. The sentence has now been rewritten:

"Using year 2100 conditions from the Representative Concentration Pathways 8.5 (RCP8.5) scenario, we quantify the individual contributions to ozone radiative forcing of (1) climate change, (2) reduced concentrations of ozone depleting substances (ODSs), and (3) methane increases. We calculate future ozone radiative forcings **and their standard error (associated with interannual variability of ozone)** relative to year 2000 of…"

**b)** Figure 3 caption: you likely mean "black dashed line" for the chemical tropopause.

**Response**: Fixed thanks.